# Autoencoding Probabilistic Circuits

**Steven Braun**[1]    **Sahil Sidheekh**[2]    **Antonio Vergari**[3]    **Martin Mundt**[4]    **Sriraam Natarajan**[2]    **Kristian Kersting**[1]

[1]Department of Computer Science, Technische Universität Darmstadt, Germany
[2]Department of Computer Science, University of Texas at Dallas, United States
[3]School of Informatics, University of Edinburgh, United Kingdom
[4]Department of Mathematics and Computer Science, Universität Bremen, Germany

## Abstract

Probabilistic circuits (PCs) are powerful probabilistic models that enable exact and tractable inference, making them highly suitable for probabilistic reasoning and inference tasks. While dominant in neural networks, representation learning with PCs remains underexplored, with prior approaches relying on external neural embeddings or activation-based encodings. To address this gap, we introduce autoencoding probabilistic circuits (APCs), a novel framework leveraging the tractability of PCs to model probabilistic embeddings explicitly. APCs extend PCs by jointly modeling data and embeddings, obtaining embedding representations through probabilistic inference. The circuit encoder is seamlessly integrated with a neural decoder in a hybrid, end-to-end trainable architecture enabled by differentiable sampling. Our empirical evaluation demonstrates that APCs outperform existing PC-based autoencoding methods in reconstruction quality, generate embeddings competitive with neural autoencoders, and exhibit superior robustness in handling missing data without the need for imputation. These results highlight APCs as a powerful and flexible representation learning method that exploits the probabilistic inference capabilities of PCs, showing promising directions for robust inference, out-of-distribution detection, and knowledge distillation.

The ability to learn compact and expressive data representations is a fundamental aspect of modern machine learning. Autoencoders have played a pivotal role in this domain by enabling the discovery of low-dimensional embeddings that capture latent data features [Hinton and Salakhutdinov, 2006]. Their impact extends across diverse applications, such as self-supervised learning paradigms like Joint Embedding Predictive Architectures [Assran et al., 2023],

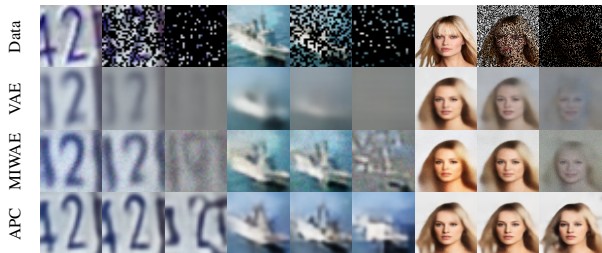

Figure 1: **APCs outperform VAE-based models in reconstruction quality under missing data.**

natural language processing, where token embeddings are crucial for Large Language Models [Vaswani et al., 2017, Devlin et al., 2019], and computer vision, where embeddings facilitate efficient image retrieval in large databases [Deng et al., 2009]. These advancements predominantly rely on neural network-based approaches, highlighting the critical role of embeddings in contemporary representation learning. Neural networks have been extensively studied in the field of representation learning, leading to their probabilistic extension, the Variational Autoencoder [Kingma and Welling, 2014, Rezende et al., 2014], along with various refinements such as vector quantization [van den Oord et al., 2017, Razavi et al., 2019], hierarchical [Vahdat and Kautz, 2020, Liévin et al., 2019] and deterministic [Ghosh et al., 2020] formulations.

In recent years, an alternative to neural networks has emerged in the form of Probabilistic Circuits (PCs; Choi et al., 2020), offering a distinct approach to probabilistic modeling. These models provide a unified framework for tractable probabilistic models [Vergari et al., 2021] and enable exact inference, addressing tractability challenges in deep generative models [Darwiche, 2003, Poon and Domingos, 2011, Kisa et al., 2014]. They have been successfully applied in diverse areas, including lossless compression [Liu et al., 2022, Severo et al., 2025], biomedical modeling [Dang et al., 2022b], neuro-symbolic AI [Ahmed et al., 2022, Loconte et al., 2023], graph representation and learning [Errica and Niepert, 2023, Zheng et al., 2018], and

*Accepted for the 8th Workshop on Tractable Probabilistic Modeling at UAI* (TPM 2025).

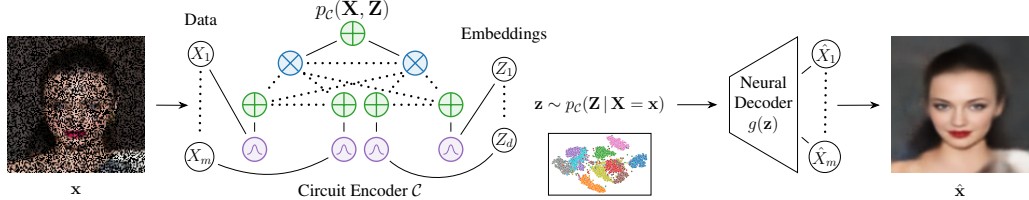

Figure 2: **Autoencoding probabilistic circuits (APCs) architecture**. An input $\mathbf{X}$ (possibly incomplete) is encoded by a probabilistic circuit $p_\mathcal{C}(\mathbf{X}, \mathbf{Z})$ into an embedding $\mathbf{z} \sim p_\mathcal{C}(\mathbf{Z} \mid \mathbf{X})$. A neural decoder reconstructs $\hat{\mathbf{X}}$ from $\mathbf{z}$.

constrained text generation [Zhang et al., 2023]. Recent advancements have enhanced PCs in expressivity, learning, and scaling through sparsity-inducing regularization [Dang et al., 2022a], structural improvements through subtractive mixtures and squared circuits [Loconte et al., 2024b,a, Wang and Broeck, 2024], as well as hybrid approaches like Hper-SPNs [Shih et al., 2021], neural conditioning in Conditional Sum-Product Networks [Shao et al., 2022] and Probabilistic Neural Circuits [Martires, 2024], and Latent Variable Distillation [Liu et al., 2023a,b]. However, in contrast to neural networks, the use of Probabilistic Circuits for representation learning has remained largely unexplored.

In this work, we present *Autoencoding Probabilistic Circuits* (APCs), a novel framework that introduces end-to-end learning of representations in an autoencoding fashion using Probabilistic Circuits. APCs leverage the probabilistic capabilities of PCs by employing them as a probabilistic encoder, generating *explicit* probabilistic embeddings $\mathbf{Z}$. Critically, embeddings in APCs are obtained through tractable conditional state inference (i.e. sampling or most probable explanation) from the joint distribution $p_\mathcal{C}(\mathbf{X}, \mathbf{Z})$ over data $\mathbf{X}$ and explicitly modeled embedding random variables $\mathbf{Z}$ modeled by the PC: $\mathbf{z} \sim p_\mathcal{C}(\mathbf{Z} \mid \mathbf{X}_o = \mathbf{x}_o)$, where $\mathbf{X}_o \subseteq \mathbf{X}$ can be partially observed, natively handling missing data (in contrast to neural based encoder, see Fig. 1). APCs then bridge probabilistic and neural architectures by allowing for an arbitrary neural network decoder. Fig. 2 illustrates the APC pipeline.

# 1 AUTOENCODING PROBABILISTIC CIRCUITS

**Probabilistic Circuits**   A *probabilistic circuit* (PC) $\mathcal{C} \equiv (\mathcal{G}, \theta)$, is a computational structure representing a function $\mathcal{C}(\mathbf{X})$ over its inputs $\mathbf{X}$, where $\mathcal{C}(\mathbf{x}) \geq 0$ for any input $\mathbf{x}$, making $\mathcal{C}(\mathbf{X})$ an unnormalized distribution. The computational graph $\mathcal{G}$, parameterized by $\theta$, consists of three types of computational units: *input* $\bigwedge$, *sum* $\bigoplus$, and *product* $\bigotimes$. An input unit $c$ encapsulates a parameterized function $f_c(\mathbf{X}_c)$, with $\mathbf{X}_c \subseteq \mathbf{X}$ denoting its scope. The scopes for product and sum units, $\mathbf{X}_c$, are the union of their input units' scopes. Specifically, a product unit computes the product of its inputs $\prod_{d \in \mathrm{IN}(c)} d(\mathbf{X}_d)$, and a sum unit computes the

weighted sum $\sum_{d \in \mathrm{IN}(c)} \theta_d^c\, d(\mathbf{X}_d)$. A PC is normalized if it fulfills two structural properties: *smoothness*, requiring every sum unit's input units to share the same scope, and *decomposability*, with product units having input units with disjoint scopes. Additionally, sum unit weights must be normalized, i.e. $\sum_{d \in \mathrm{IN}(c)} \theta_d^c = 1$.

We now introduce Autoencoding Probabilistic Circuits (APCs), a novel approach to autoencode data in probabilistic circuits that leverages their tractable inference capabilities. In contrast to CAT and ACT embeddings introduced in Vergari et al. [2018], we propose to model embedding random variables with *explicit* distributions represented by additional input units in the PC structure. This enables us to perform probabilistic inference such as sampling and most probable explanation (MPE) assignments to obtain and reason about embeddings, providing a more principled embedding representation.

APCs extend the set of data variables $\mathbf{X}$ by an additional set of embedding variables $\mathbf{Z}$ that encode a compressed representation of $\mathbf{X}$. As is common for autoencoder models, APCs define two essential components: (1) the encoder $f : \mathbf{X} \mapsto \mathbf{Z}$, mapping data samples $\mathbf{x} \in \mathbf{X}$ to their respective embedding representations $\mathbf{z} \in \mathbf{Z}$, and (2) the decoder $g : \mathbf{Z} \mapsto \mathbf{X}$, mapping a given embedding $\mathbf{z}$ to data reconstructions $\hat{\mathbf{x}}$. Consequently, applying the decoder to the encoder will result in an approximate reconstruction of the original data sample $\hat{\mathbf{x}} \approx g(f(\mathbf{x}))$. In the following, we will discuss the encoding and decoding procedures in detail.

## 1.1   ENCODING

As the encoder model, we construct a smooth and decomposable Probabilistic Circuits $\mathcal{C}$ that models the joint distribution $p_\mathcal{C}(\mathbf{X}, \mathbf{Z})$ of data and embedding random variables. The encoding process $f_\mathcal{C} : \mathbf{X} \mapsto \mathbf{Z}$ maps a data sample $\mathbf{x}$ to its embedding representation $\mathbf{z}$. We define the encoding $f_\mathcal{C}(\mathbf{x})$ as performing probabilistic state inference in the PC by conditionally sampling an embedding, given the data $\mathbf{x}$

$$\mathbf{z} \sim p_\mathcal{C}(\mathbf{Z} \mid \mathbf{X} = \mathbf{x}) . \qquad (1)$$

Conditional sampling in PCs consists of two passes through the network. First, a bottom-up pass of the marginal $p_\mathcal{C}(\mathbf{X} = \mathbf{x})$ for a given data sample $\mathbf{x}$ caches log-likelihoods

at each input edge. Subsequently, a top-down pass with reweighted sum unit weights according to their cached log-likelihoods samples from the on $\mathbf{X} = \mathbf{x}$ conditioned marginal $p_{\mathcal{C}|_{\mathbf{x}=\mathbf{x}}}(\mathbf{Z}) = p_{\mathcal{C}}(\mathbf{Z}\,|\,\mathbf{X} = \mathbf{x})$. When choosing a PC that is smooth and decomposable, Eq. (1) becomes tractable.

Moreover, and in stark contrast to neural encoders, a PC-based encoding scheme allows us to infer the full embedding state $\mathbf{z}$ from *partial evidence*, where only a subset $\mathbf{X}_o \subseteq \mathbf{X}$ of the random variables are observed and the rest is missing $\mathbf{X}_m = \mathbf{X} \setminus \mathbf{X}_o$. This is achieved through marginalizing the missing variables $\int p_{\mathcal{C}}(\mathbf{X}_o, \mathbf{X}_m)\, d\mathbf{X}_m$, without the need for any data imputation. On the other hand, neural networks assume all input values to be given. Consequently, data with missing observations needs to be manually preprocessed before being fed to a neural network. When not natively accounted for in a specific method, this preprocessing typically involves employing various imputation techniques such as mean imputation, multiple imputation [Rubin, 1989], or advanced methods like Expectation-Maximization algorithms and deep learning approaches [Yoon et al., 2018, Gondara and Wang, 2018, Luo et al., 2018]. As a side effect, for neural networks these imputation methods usually introduce biases or assumptions that can affect models.

Lastly, for the embedding $\mathbf{z}$ to be used in the autoencoding pipeline and training procedure of the model, we need the encoding function $f$ to be differentiable. Here, we make use of the concept of differentiable sampling for Probabilistic Circuits introduced in Lang et al. [2022]. This allows us to replace the discrete operations present in the sampling procedure of sum units with continuous, and thus differentiable, reparametrizations. We further improve upon the method proposed in Lang et al. [2022] and replace the Gumbel-Softmax trick with SIMPLE [Ahmed et al., 2023]. SIMPLE is a gradient estimator for $k$-subset sampling that uses discrete sampling in the forward pass and exact conditional marginals in the backward pass. This approach maintains discreteness during forward propagation while efficiently computing exact marginals for gradient estimation during backpropagation. By incorporating SIMPLE into the differentiable sampling procedure for Probabilistic Circuits, we are able to improve gradient estimation and overall performance of the autoencoding pipeline, overcoming the initial difficulties of differentiable sampling in PCs reported in Lang et al. [2022], which we show in Appendix I.

While not explicitly explored in this work, we emphasize that the use of a circuit encoder enables leveraging data-specific input or even embedding distributions to accurately model random variables (RVs). This is particularly beneficial in scenarios where we have prior knowledge about the specific distributions from which RVs are drawn, or when each RV can be effectively represented as a mixture of multiple distributions of different types.

## 1.2 DECODING

In Autoencoding Probabilistic Circuits, while the encoding phase leverages the principled and tractable nature of Probabilistic Circuits, the decoding process employs a neural network to reconstruct data from the learned embeddings. This design choice is motivated by the complementary strengths of neural networks, particularly their capacity for modeling complex, non-linear mappings. Therefore, in APCs, the decoder $g\colon \mathbf{Z} \mapsto \mathbf{X}$ is implemented as a separate neural network, distinct from the probabilistic circuit encoder $\mathcal{C}$. This neural decoder, maps the embedding representation $\mathbf{z}$ back to the original data space $\mathbf{X}$, effectively complementing the probabilistic encoding with the modeling capacity and computational efficiency of neural networks during decoding. The specific architecture of the neural decoder can be tailored to the data type; for instance, CNNs can be employed to capture spatial relationships in image data, allowing for the incorporation of implicit biases beneficial for reconstruction that are not explicitly modeled in the encoding circuit.

Importantly, this neural decoding strategy remains robust even when dealing with partial evidence. The probabilistic encoder $f_{\mathcal{C}}$'s ability to infer a complete embedding $\mathbf{z}$ from a partially observed input $\mathbf{x}_o$ ensures that the neural decoder always receives a full embedding vector. Consequently, the decoding process is unaffected by missing data in the input. We empirically explore the benefits of this hybrid approach by comparing APCs with neural decoders to APCs with self-decoding Probabilistic Circuits in Appendix B.6.

## 1.3 END-TO-END TRAINING

Autoencoding Probabilistic Circuits are trained end-to-end using objectives that encourage both accurate reconstruction and meaningful and robust embedding representations. The loss function is a sum of three crucial components, which we will now describe in detail.

**Embedding Regularization** To ensure the learned embeddings are meaningful and to avoid overfitting, we constrain the embedding distribution to a chosen prior. While simple $L_p$ regularization could be applied to the embedding vectors $\mathbf{z}$, Probabilistic Circuits allow for a more principled approach. Sampling in a Probabilistic Circuit induces a product unit tree with independent input units. Consider a simple circuit with a sum unit acting as a mixture over two independently modeled random variables A and B:

$$p_{\mathcal{C}}(\mathrm{A}, \mathrm{B}) = w_1 p_{\mathrm{A}}^1(\mathrm{A})\, p_{\mathrm{B}}^1(\mathrm{B}) + w_2 p_{\mathrm{A}}^2(\mathrm{A})\, p_{\mathrm{B}}^2(\mathrm{B}) \quad . \quad (2)$$

Sampling from $p_{\mathcal{C}}$ selects one mixture component $i \in 1, 2$, resulting in input units $p_{\mathrm{A}}^i(\mathrm{A})$ and $p_{\mathrm{B}}^i(\mathrm{B})$ that are independent in this circuit structure. Generalizing this concept to our encoder, sampling embeddings from the conditional distribution $p_{\mathcal{C}}(\mathbf{Z}\,|\,\mathbf{X})$ induces a tree $\mathcal{C}'$ in which embeddings

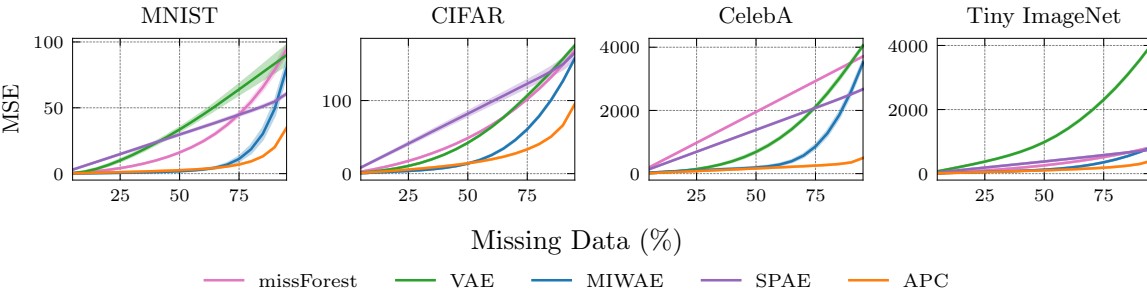

Figure 3: **APCs can deliver lower reconstruction errors than PC, VAE, and missForest baselines in the asymptotic regime of MCAR-style randomly missing data**. Reconstruction mean squared error is reported across all image datasets as the degree of MCAR corruption increases from 0% to 95%.

are mutually conditionally independent, given $\mathbf{x}$:

$$Z_j \perp Z_k \,|\, \mathbf{X}, \quad \forall j, k \in 1, \ldots, |\mathbf{Z}|, j \neq k \quad . \quad (3)$$

This independence allows us to apply closed-form statistical distances between the induced embedding distribution and a chosen prior $q$. Specifically, we use the Kullback-Leibler divergence (KLD):

$$\mathcal{L}_{\text{KLD}} = \sum_{i=1}^{B} \sum_{j=1}^{|\mathbf{Z}|} \text{KLD}\big(p_{\mathcal{C}'}(Z_j \,|\, \mathbf{X} = \mathbf{x}_i) \,||\, q(Z_j)\big) \quad (4)$$

**Joint Data and Embedding Likelihood**   Using a PC as an encoder further offers the advantage of modeling the joint distribution of data and embedding spaces in a probabilistic manner, allowing us to directly maximize their joint likelihood. The maximum likelihood estimation (MLE) using the negative log-likelihood loss function is defined as follows:

$$\mathcal{L}_{\text{NLL}} = -\frac{1}{B} \sum_{i=1}^{B} \log p_{\mathcal{C}}(\mathbf{x}_i, \mathbf{z}_i) \quad , \quad (5)$$

where $B$ is the batch size, $\mathbf{x}_i$ represents the $i$-th data sample, and $\mathbf{z}_i$ is the corresponding embedding.

**Reconstruction**   Similar to fully neural autoencoders, we use a reconstruction term $\mathcal{L}_{\text{rec}}$ to ensure that embeddings represent sufficient information to accurately reconstruct the original input data, thereby encouraging the model to learn a meaningful and compact embedding space that captures essential features of the data generating distribution:

$$\mathcal{L}_{\text{rec}}(\mathbf{x}) = -\frac{1}{B} \sum_{i=1}^{B} \log p_\theta(\mathbf{x}_i \,|\, \mathbf{z}_i), \quad \mathbf{z}_i \sim p_{\mathcal{C}}(\mathbf{Z} \,|\, \mathbf{x}_i), \quad (6)$$

where $B$ is the batch size, $D$ is the dimension of the input data, $\mathbf{x}_i$ is the $i$-th input sample, and $\hat{\mathbf{x}}_i = g_\theta(f_{\mathcal{C}}(\mathbf{x}_i))$ is its corresponding reconstruction.

## 2   EVALUATION

We evaluate APCs' reconstructions on image (Fig. 3 and Appendix Table 1) and tabular (Appendix Table 2) data under increasing missing completely at random (MCAR) style pixel corruption. We compare APCs against Sum-Product Autoencoding (SPAE) [Vergari et al., 2018], vanilla Variational Autoencoder (VAE) [Kingma and Welling, 2014], a missing-data specific VAE variant (MIWAE) [Mattei and Frellsen, 2019], and missForest [Stekhoven and Bühlmann, 2011], a classic machine learning approach. APCs consistently demonstrate superior performance across all datasets, achieving lower average Mean Squared Error (MSE) compared to both neural, PC-based and the classic baseline. We provide additional details and qualitative and quantitative results in Appendices B and D to I.

## 3   CONCLUSION

In this work, we introduced Autoencoding Probabilistic Circuits (APCs), a novel framework where a Probabilistic Circuit (PC) encoder explicitly models probabilistic embeddings by jointly representing data and latent variables. This enables tractable conditional sampling for encoding and direct inference on embeddings, contrasting with often intractable formulations like Variational Autoencoders. Our hybrid architecture pairs this PC encoder with a neural decoder, trained end-to-end. Empirical evaluations across diverse datasets demonstrated APCs' effectiveness, including superior reconstruction performance over existing circuit-based and neural encoder methods, particularly in the presence of increasing data corruption and missing values. The learned embeddings proved robust and meaningful, maintaining high downstream task performance even under significant data corruption. We further highlighted APCs' generative capabilities and their potential for out-of-distribution detection and data-free knowledge distillation, stemming from their explicit probabilistic nature and tractable marginalization. We outline future work in Appendix J.

## Acknowledgements

This work was supported by the Federal Ministry of Education and Research (BMBF) Competence Center for AI and Labour ("kompAKI", FKZ 02L19C150) and benefited from the Hessian Ministry of Higher Education, Research, Science and the Arts cluster projects "The Third Wave of AI" as well as the preparation of the "Reasonable AI" Cluster of Excellence that will be funded by the Deutsche Forschungsgemeinschaft (DFG, German Research Foundation) under Germany's Excellence Strategy – EXC 3057/1. AV was supported by the "UNREAL: A Unified Reasoning Layer for Trustworthy ML" project (EP/Y023838/1) selected by the ERC and funded by UKRI EPSRC. SS and SN gratefully acknowledge the generous support by the AFOSR award FA9550-23-1-0239 and the DARPA Assured Neuro Symbolic Learning and Reasoning (ANSR) award HR001122S0039.

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

# Autoencoding Probabilistic Circuits
## (Supplementary Material)

**Steven Braun**[1]   **Sahil Sidheekh**[2]   **Antonio Vergari**[3]   **Martin Mundt**[4]   **Sriraam Natarajan**[2]   **Kristian Kersting**[1]

[1]Department of Computer Science, Technische Universität Darmstadt, Germany
[2]Department of Computer Science, University of Texas at Dallas, United States
[3]School of Informatics, University of Edinburgh, United Kingdom
[4]Department of Mathematics and Computer Science, Universität Bremen, Germany

## A   RELATED WORK

Having introduced Autoencoding Probabilistic Circuits (APCs) in the main body, we now contextualize our framework by discussing prior work in representation learning using both Probabilistic Circuits and neural networks and prior hybrid approach between these two classes of models.

**Representation Learning with Probabilistic Circuits**   While Probabilistic Circuits (PCs) have seen significant advancements in expressivity and learning algorithms [Liu et al., 2023a,b, Dang et al., 2022a, Ventola et al., 2023, Trapp et al., 2019, Yang et al., 2023, Gala et al., 2024a,b, Loconte et al., 2024b, Wang and Broeck, 2024], their application to representation learning in an autoencoding fashion has been comparatively limited. The main work in this area is Sum-Product Autoencoding (SPAE; Vergari et al., 2018), which proposed two encoding strategies for Sum-Product Networks: CAT embeddings, derived from MPE inference over latent variables, and ACT embeddings, using circuit activations as continuous representations. However, SPAE derives these embeddings post-hoc and does not incorporate explicit embedding learning into its training process. In Appendix F we additionally evaluate the APC framework with SPAE encoder and decoder, and a neural decoder. Other research has explored integrating PCs with deep generative models (DGMs). This includes using neural networks to guide PC structure learning or act as conditional components [Shih et al., 2021, Shao et al., 2022, Martires, 2024], incorporating deep learning inductive biases into PCs [Ventola et al., 2020, Butz et al., 2019, Yu et al., 2022], or using PCs within hybrid DGMs for tasks like knowledge distillation or in combination with normalizing flows and VAEs [Liu et al., 2023a,b, Pevný et al., 2020, Sidheekh et al., 2023, Correia et al., 2023, Gala et al., 2024a]. Dennis and Ventura [2017] proposed a hybrid model using two separate circuits for data and embeddings alongside a neural autoencoder to refine samples. In contrast to these approaches, APCs introduce a framework where probabilistic embeddings are *explicitly* modeled as random variables within a single PC encoder and are learned end-to-end.

**Neural Autoencoders for Representation Learning**   Neural autoencoders (AEs) form the foundation of many modern representation learning techniques [Hinton and Salakhutdinov, 2006]. Early extensions like Denoising Autoencoders (DAEs; Vincent et al., 2008) and Contractive Autoencoders (CAEs; Salah et al., 2011) focused on learning robust and invariant features. The advent of Variational Autoencoders (VAEs; Kingma and Welling, 2014, Rezende et al., 2014) introduced a principled probabilistic approach, enabling generative modeling by learning a stochastic mapping to a latent space, typically regularized towards a simple prior. Despite their success, VAEs often rely on approximate inference and simplified posterior distributions (e.g., Gaussians), which can limit their expressiveness. Significant research has aimed to overcome these limitations through richer posterior families like normalizing flows [Rezende and Mohamed, 2015, Kingma et al., 2016, Louizos and Welling, 2016], hierarchical latent structures [Sønderby et al., 2016, Vahdat and Kautz, 2020], discrete representations like VQ-VAEs [van den Oord et al., 2017, Razavi et al., 2019], and improved training objectives [Burda et al., 2016, Higgins et al., 2017, Tomczak and Welling, 2018]. Deterministic variants like Regularized Autoencoders (RAEs; Ghosh et al., 2020) have also been explored to structure the latent space without the sampling complexities of VAEs, often relying on post-hoc density estimation for generation. Furthermore, to address the real-world scenario of missing data, various neural AE adaptations have been proposed. These include techniques such as importance-weighted training (MIWAE; Mattei and Frellsen, 2019), tailored likelihoods for heterogeneous missing data (HI-VAE; Nazabal et al., 2020), modified

*Accepted for the 8th Workshop on Tractable Probabilistic Modeling at UAI*  (TPM 2025).

losses in DAEs (mDAE; Dupuy et al., 2024), and advanced inference or sampling strategies in hierarchical or iterative models [Peis et al., 2022, Kuang et al., 2024]. While powerful, these neural autoencoding paradigms, even those specialized for missing data, often involve approximate inference for their probabilistic embeddings or necessitate imputation strategies for incomplete data. APCs, by contrast, leverage a PC encoder for exact probabilistic inference over embeddings and an intrinsic capability to handle missing observations via marginalization. The empirical comparison in Appendix B will further substantiate these distinctions.

Table 1: Average reconstruction performance on eight image datasets under increasing percentages of MCAR-style randomly missing pixels (0%-95%). APCs consistently outperform all other methods in maintaining reconstruction quality as the proportion of missing data increases.

| | APC | SPAE | VAE | MIWAE | missForest |
|---|---|---|---|---|---|
| MNIST [LeCun et al., 1998] | **5.07** $_{\pm 0.04}$ | 28.46 $_{\pm 0.43}$ | 35.38 $_{\pm 1.74}$ | 9.16 $_{\pm 0.92}$ | 24.51 $_{\pm 0.63}$ |
| F-MNIST [Xiao et al., 2017] | **4.33** $_{\pm 0.02}$ | 18.96 $_{\pm 0.11}$ | 39.80 $_{\pm 1.10}$ | 12.54 $_{\pm 0.74}$ | 24.53 $_{\pm 0.02}$ |
| CIFAR [Krizhevsky, 2009] | **20.65** $_{\pm 0.09}$ | 78.32 $_{\pm 3.09}$ | 55.81 $_{\pm 0.47}$ | 32.80 $_{\pm 0.27}$ | 57.61 $_{\pm 0.16}$ |
| CelebA [Liu et al., 2015] | **166.89** $_{\pm 0.65}$ | 1318.73 $_{\pm 6.37}$ | 1095.90 $_{\pm 14.0}$ | 576.47 $_{\pm 24.8}$ | 1855.67 $_{\pm 2.29}$ |
| SVHN [Netzer et al., 2011] | **4.87** $_{\pm 0.00}$ | 35.38 $_{\pm 1.92}$ | 41.01 $_{\pm 0.15}$ | 17.28 $_{\pm 0.29}$ | 44.19 $_{\pm 0.25}$ |
| Flowers [Gurnani et al., 2017] | **45.22** $_{\pm 0.14}$ | 119.31 $_{\pm 5.23}$ | 107.75 $_{\pm 11.8}$ | 58.03 $_{\pm 0.37}$ | 52.65 $_{\pm 0.14}$ |
| LSUN [Yu et al., 2015] | **63.70** $_{\pm 0.20}$ | 322.22 $_{\pm 3.48}$ | 254.75 $_{\pm 2.39}$ | 123.58 $_{\pm 1.40}$ | 115.28 $_{\pm 0.15}$ |
| ImageNet [Deng et al., 2009] | **118.70** $_{\pm 0.16}$ | 365.06 $_{\pm 2.20}$ | 1286.51 $_{\pm 10.4}$ | 204.10 $_{\pm 2.04}$ | 295.40 $_{\pm 0.34}$ |

# B  EMPIRICAL EVALUATION

To measure the effectiveness of APCs, we conduct a comprehensive series of experiments designed to evaluate their performance and robustness across multiple tests against an array of model choices. We include the existing autoencoding-mechanism SPAE introduced in Vergari et al. [2018] as a PC-based reference, a vanilla Variational Autoencoder [Kingma and Welling, 2014] as a neural autoencoding baseline, MIWAE [Mattei and Frellsen, 2019] as a VAE variant specifically designed for handling missing data through multiple imputation rounds, and missForest [Stekhoven and Bühlmann, 2011], an iterative imputation method based on random forests as a strong classic machine learning baseline. We repeat all experiments five times with different seeds and report their mean and standard deviation. All models are evaluated on common image benchmark datasets, as well as the 20 DEBD binary tabular datasets [Lowd and Davis, 2010, Haaren and Davis, 2012, Bekker et al., 2015, Larochelle and Murray, 2011]. To assess robustness, we evaluate models under two primary missing data mechanisms: Missing Completely At Random (MCAR) and Missing At Random (MAR). For MCAR, we introduce increasing levels of input corruption by randomly dropping pixels chosen from a uniform distribution, with the percentage of missing data ranging from 0% to 95% in 5% steps. This is the default corruption method unless otherwise specified. For MAR, we evaluate specific structured missingness patterns, such as missing entire regions of an image (e.g., left-to-right, center-to-border). For each metric, we report the missing-input mean squared reconstruction error (MSE) averaged over the different levels of corruptions. For additional details, we refer the reader to Appendix D. Notably, missForest is an imputation algorithm that operates directly in the input space, unlike the autoencoding methods which learn a compressed representation in an embedding space. Consequently, when no data is missing (0% corruption), missForest can achieve a perfect reconstruction error of 0.0, whereas autoencoder-based approaches are inherently limited by their embedding bottleneck, resulting in non-zero reconstruction error even with complete inputs.

## B.1  RECONSTRUCTIONS: COMPARING CIRCUIT AND NEURAL ENCODERS

We first evaluate the reconstruction capabilities of APCs and examine performance on image data in Appendix Table 1 (and tabular data in Appendix Table 2), with attention to how well models handle increasing levels of data corruptions. Appendix Table 1 presents the reconstruction performance of all models on image datasets with progressively increasing percentages of randomly missing pixels, reported as average mean squared error (and Structural Similarity Index Measure (SSIM) in Appendix Table 5), capturing reconstruction quality over differently magnitudes of corruption. It is important to highlight, that with the exception of SPAE, all other models utilize *the same neural decoder architecture*. Therefore, the observed differences in performance can be solely attributed to the choice of the encoder model and the corresponding encoding scheme across the bench. APCs demonstrate superior performance across all datasets, with lower MSE and higher SSIM compared to both neural and PC-based baselines.

To visualize this robustness advantage, Appendix Figure 4 shows reconstruction error trends as the percentage of MCAR-style missing pixels increases from 0% to 95%. While all models perform comparably at low corruption levels, their results diverge quickly as corruption increases: neural autoencoders show rapid performance degradation, while APCs maintain lower reconstruction error even at high corruption levels. This pattern is consistent across all datasets. In addition, qualitative examples of reconstructions with missing data in Appendix Figure 5 provide visual confirmation of our quantitative findings.

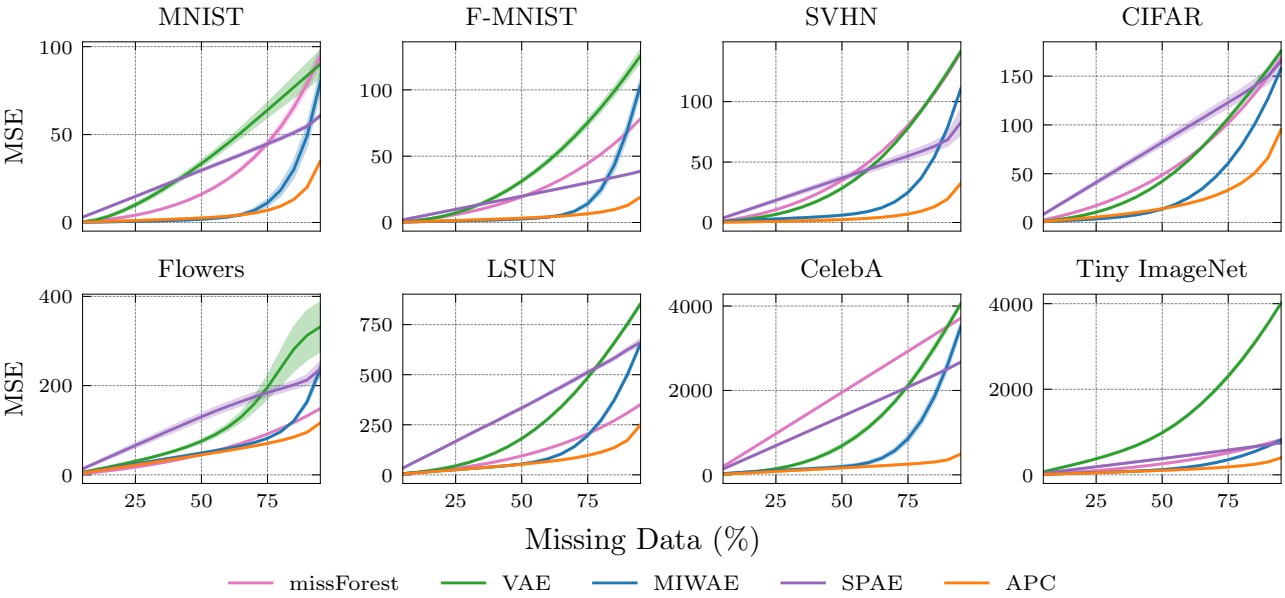

Figure 4: Reconstruction mean squared error on all image datasets for increasing degrees of MCAR-style randomly missing data (0% - 95%). APCs are able to outperform all other models by large margins on all datasets with increasing fractions of missing data.

We show reconstructions for 0%, 50%, 80% MCAR-style missing data, and vertical-band as well as center-square MAR-style missing data. As missing data increases, neural autoencoders produce increasingly blurry and eventually unrecognizable reconstructions. In contrast, APCs maintain recognizable reconstructions even at high levels of missing data or images missing complete patches, preserving key structural elements and details across all datasets. We attribute this robustness to the encoder circuit's ability to natively handle partial observations through tractable marginalization.

While image data inherently contains spatial structure and correlations that neural architectures can exploit through layers with implicit biases such as convolutions, tabular data typically lacks such inherent structure. We therefore additionally investigate whether APCs' strong performance on image datasets generalizes to tabular data. Appendix Table 2 presents reconstruction performance on the 20 DEBD binary tabular datasets under increasing percentages of missing values. APCs outperform all other models on 18 out of 20 datasets, demonstrating that their advantages are not limited to image data but generalize to tabular datasets. This is particularly significant for tabular data applications where missing values are common in real-world scenarios, such as medical records, survey and financial data or environmental modeling.

## B.2 EMBEDDING QUALITY AND DOWNSTREAM TASK PERFORMANCE

While quantitative and qualitative experiments in Appendix B.1 so far have investigated reconstruction performance, a key indicator of representation learning quality is how well embeddings can be used for downstream task applications after *unsupervised pre-training* of the autoencoding model. For this, a logistic regression model was trained iteratively using stochastic gradient descent (SGD), for details, we refer the reader to Appendix D. By utilizing a logistic regression model, we aim to evaluate the learned embeddings' ability to capture independent and useful data representations. The downstream task performance is measured by the downstream task classification accuracy (DS-Acc. ↑), providing insights into the quality and linear separability of the learned representations. Appendix Figure 6 presents downstream classification accuracy of the logistic regression model trained on embeddings extracted from four image datasets under increasing levels of MCAR-style corruptions. The results reveal a stark contrast between APCs and neural autoencoder variants: while neural encoders achieve slightly higher accuracy with complete data (0% missing), APC embeddings maintain their classification performance even as corruption increases to severe levels (90%), whereas neural encoder embeddings show immediate degradation in downstream performance, quickly converging towards a baseline random guessing estimator, when data is missing. This pattern is consistent across all datasets. The robustness to increasing proportions of missing data can again be attributed to the circuit's fundamental capability for tractable marginalization, allowing inference of embeddings even with

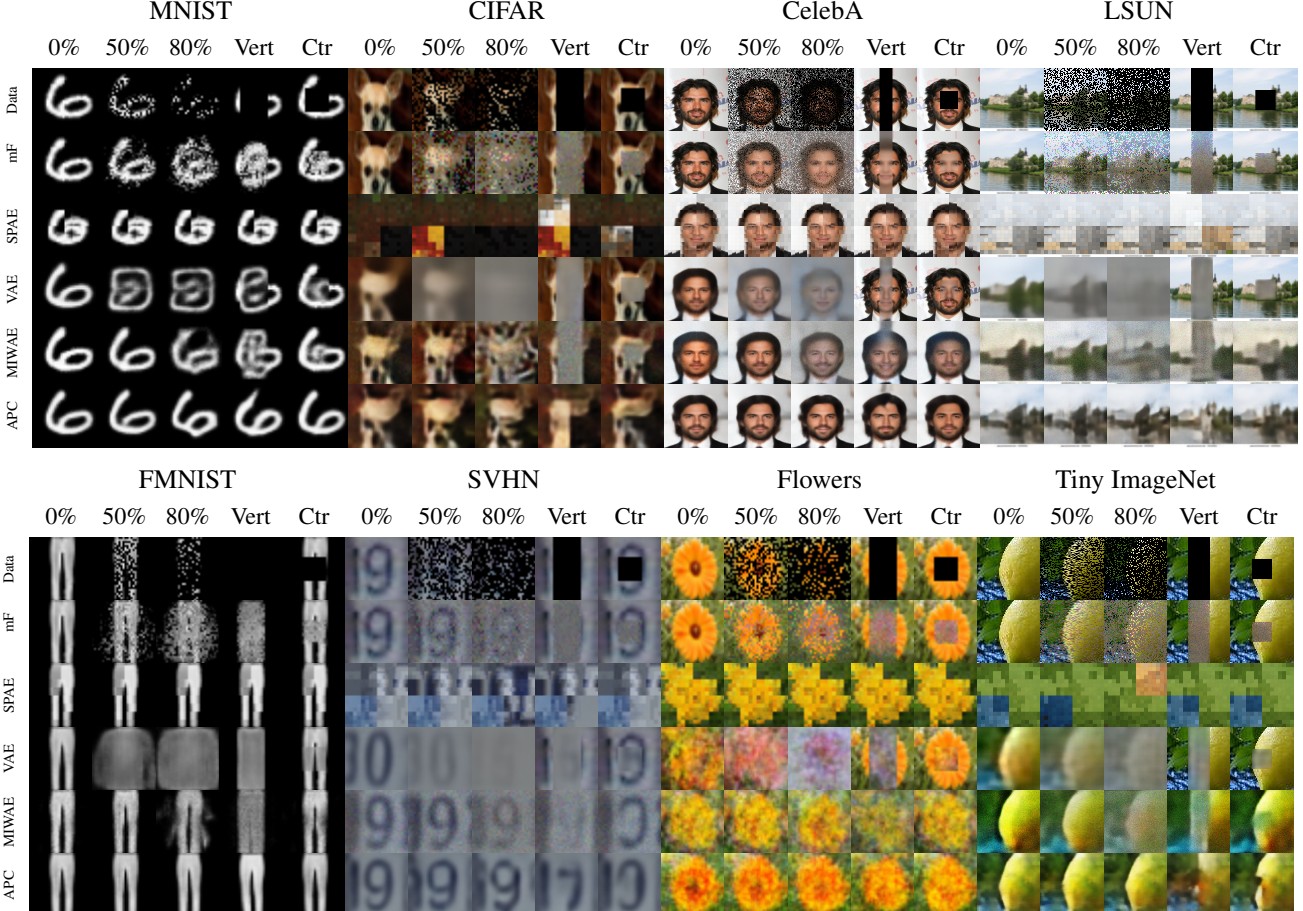

Figure 5: Exemplary image dataset reconstructions under different MAR and MCAR corruptions. The first row (Data) shows inputs with different corruptions, and subsequent rows represent the respective model reconstructions. While neural encoder models quickly collapse with an increase in missing data ratio and mostly fail on MAR corruptions, APCs are able to maintain a stable reconstruction, even at high degrees of corruption.

incomplete observations without the need for additional imputation methods.

To provide visual insight into these performance differences, Appendix Figure 7 shows t-SNE projections of embedding spaces for the MNIST test dataset split at different corruption levels. Because SPAE-ACT embeddings represent circuit activations, which are log-likelihoods and can have large negative values, we normalize them prior to t-SNE to ensure a comparable input range with other models. With complete data (0% corruption), both neural autoencoders and APCs produce well-structured embeddings with clear class separation. However, as corruption increases, neural encoder, even MIWAE which is specifically trained on missing data, embeddings progressively collapse into an unstructured mass with no class separation. In contrast, APC embeddings maintain their structure with distinct class clusters even at 90% corruption, visually confirming our quantitative results of Appendix Figure 6 and their robustness to missing data.

## B.3 EMBEDDING SPACE ANALYSIS

A key advantage of APCs is the fact, that the encoder models the joint data-embedding distribution $p_{\mathcal{C}}(\mathbf{X}, \mathbf{Z})$ explicitly and is able to tractable compute marginals, enabling additional probabilistic capabilities compared to standard and Variational Autoencoders. In this section, we briefly examine APCs from a probabilistic perspective by investigating the embedding space and samples decoded from the embedding space.

Unlike VAEs, which implicitly model the approximate posterior distribution $q(\mathbf{Z} \mid \mathbf{X})$ using a neural network and rely on approximate inference, APCs directly model the joint distribution $p_{\mathcal{C}}(\mathbf{X}, \mathbf{Z})$ and allow for exact and tractable probabilistic queries. VAEs are trained to encourage the encoder's approximate posterior $q(\mathbf{Z} \mid \mathbf{X})$ to approach a prior distribution,

Table 2: Reconstruction performance on the 20 DEBD binary tabular datasets under increasing percentages of MCAR-style randomly missing data (0%-95%). APCs achieve the best reconstruction performance on 18 of the 20 datasets.

| | APC | SPAE | VAE | MIWAE | missForest |
|---|---|---|---|---|---|
| accidents | **6.16** $\pm 0.04$ | 12.55 $\pm 0.86$ | 7.56 $\pm 0.28$ | 7.34 $\pm 0.05$ | 6.75 $\pm 0.01$ |
| ad | 5.57 $\pm 0.03$ | 231.84 $\pm 15.4$ | 6.04 $\pm 0.17$ | **3.50** $\pm 0.06$ | 5.57 $\pm 0.03$ |
| baudio | **6.50** $\pm 0.03$ | 15.60 $\pm 1.33$ | 7.44 $\pm 0.01$ | 7.81 $\pm 0.04$ | 7.50 $\pm 0.01$ |
| bbc | **34.66** $\pm 0.15$ | 161.59 $\pm 10.8$ | 37.45 $\pm 0.55$ | 47.08 $\pm 0.60$ | 35.02 $\pm 0.15$ |
| bnetflix | **10.08** $\pm 0.02$ | 20.53 $\pm 0.96$ | 13.49 $\pm 0.77$ | 14.81 $\pm 0.04$ | 10.80 $\pm 0.00$ |
| book | **3.65** $\pm 0.02$ | 67.10 $\pm 9.40$ | 3.73 $\pm 0.02$ | 4.40 $\pm 0.06$ | 3.87 $\pm 0.03$ |
| c20ng | **19.12** $\pm 0.04$ | 123.50 $\pm 5.85$ | 20.69 $\pm 0.05$ | 25.50 $\pm 0.28$ | 20.34 $\pm 0.02$ |
| cr52 | **11.94** $\pm 0.10$ | 134.30 $\pm 7.81$ | 12.83 $\pm 0.18$ | 15.10 $\pm 0.19$ | 12.75 $\pm 0.09$ |
| cwebkb | **21.32** $\pm 0.14$ | 109.07 $\pm 4.38$ | 22.94 $\pm 0.15$ | 28.53 $\pm 0.32$ | 22.71 $\pm 0.07$ |
| dna | **15.89** $\pm 0.12$ | 28.76 $\pm 0.91$ | 16.46 $\pm 0.17$ | 20.46 $\pm 0.30$ | **15.89** $\pm 0.04$ |
| jester | **9.04** $\pm 0.02$ | 20.98 $\pm 0.33$ | 17.44 $\pm 0.10$ | 15.04 $\pm 0.08$ | 10.66 $\pm 0.01$ |
| kdd | **0.19** $\pm 0.00$ | 4.12 $\pm 0.57$ | 0.19 $\pm 0.00$ | 0.22 $\pm 0.00$ | 0.20 $\pm 0.00$ |
| kosarek | **1.35** $\pm 0.04$ | 13.57 $\pm 3.63$ | 1.42 $\pm 0.01$ | 1.67 $\pm 0.01$ | 1.40 $\pm 0.00$ |
| moviereview | **48.06** $\pm 0.15$ | 141.19 $\pm 4.27$ | 50.08 $\pm 0.22$ | 70.07 $\pm 0.13$ | 48.36 $\pm 0.06$ |
| msnbc | 1.05 $\pm 0.03$ | 2.23 $\pm 0.39$ | 1.03 $\pm 0.01$ | 1.17 $\pm 0.00$ | **0.99** $\pm 0.00$ |
| nltcs | **1.12** $\pm 0.02$ | 2.48 $\pm 0.00$ | 1.60 $\pm 0.01$ | 1.37 $\pm 0.01$ | 1.48 $\pm 0.00$ |
| plants | **2.52** $\pm 0.04$ | 11.62 $\pm 0.54$ | 3.93 $\pm 0.05$ | 2.93 $\pm 0.01$ | 4.67 $\pm 0.01$ |
| pumsb_star | **5.82** $\pm 0.16$ | 23.30 $\pm 0.74$ | 12.47 $\pm 0.40$ | 6.53 $\pm 0.10$ | 11.74 $\pm 0.02$ |
| tmovie | **7.32** $\pm 0.06$ | 57.94 $\pm 5.49$ | 8.72 $\pm 0.03$ | 9.04 $\pm 0.04$ | 10.11 $\pm 0.02$ |
| tretail | **1.22** $\pm 0.01$ | 6.22 $\pm 1.63$ | 1.23 $\pm 0.00$ | 1.58 $\pm 0.02$ | **1.22** $\pm 0.00$ |
| voting | **27.71** $\pm 0.18$ | 264.53 $\pm 7.30$ | 122.24 $\pm 52.8$ | 39.04 $\pm 0.46$ | 119.03 $\pm 0.12$ |

typically a standard normal distribution $\mathcal{N}(0, 1)$. This is achieved by incorporating a KLD term in the loss function that measures the discrepancy between $q(\mathbf{Z} \mid \mathbf{X})$ and $\mathcal{N}(0, 1)$. Samples are drawn from $\mathcal{N}(0, 1)$ after training, assuming the learned approximate posterior is a good approximation of the true posterior $p(\mathbf{Z} \mid \mathbf{X})$. However, the encoder rarely perfectly matches the true posterior, leading to a mismatch between the true posterior and the distribution used for sampling. In contrast, APCs allow us to sample embeddings directly from the exact marginal distribution $\mathbf{z} \sim p_{\mathcal{C}}(\mathbf{Z})$, obtained tractably through marginalization of the probabilistic circuit: $\int p_{\mathcal{C}}(\mathbf{x}, \mathbf{Z}) \, d\mathbf{x}$. These embeddings are then decoded to generate new data samples. Appendix Figure 8 showcases these decoded embedding samples across datasets of varying complexity and contrasts them with a PC solely trained with MLE. The generated samples exhibit diversity and maintain the structural characteristics of their respective datasets, ranging from clear digit forms in MNIST to more complex architectural features in LSUN and facial attributes in CelebA. Vanilla PC samples exhibit well known visual artifacts that are related to the circuit structure. APCs are able to circumvent this by deferring the decoding to a neural network. This demonstrates that APCs successfully capture the underlying data distribution through their probabilistic encoder, generating higher quality samples by decoding from the embedding distribution and natively avoiding posterior approximation discrepancies inherent in VAEs due to the intractable nature of their posterior distributions.

## B.4 MODEL CAPACITY IS INSUFFICIENT FOR MISSING DATA ROBUSTNESS EVEN IN STATE-OF-THE-ART VAES

Our preceding analyses have primarily focused on neural encoder and decoder architectures. However, the autoencoding and in particular the VAE landscape has witnessed significant advancements, leading to more sophisticated and complex architectures as well as training procedures. Notable examples include Vector Quantized VAEs (VQ-VAE) [van den Oord et al., 2017] and VQ-VAE2 [Razavi et al., 2019], as well as very deep models like VDVAE [Child, 2020] and hierarchical ones like NVAE [Vahdat and Kautz, 2020]. To find out whether robust reconstruction under missing data is predominantly a function of model scale, complexity, and capacity, we evaluate NVAE under MAR and MCAR corruptions as a representative for modern VAE architectures.

We employed the official publicly available pretrained NVAE models for CIFAR and CelebA from Vahdat and Kautz

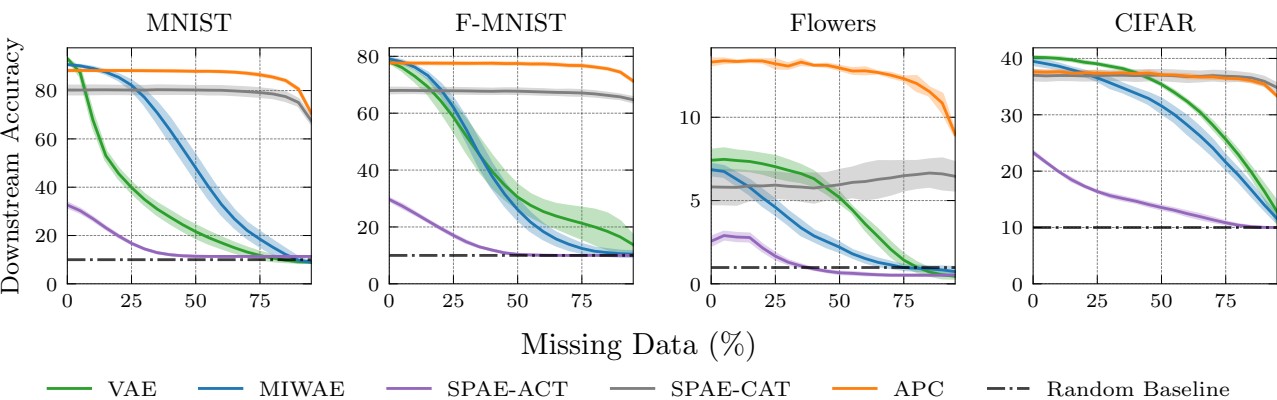

Figure 6: Downstream task accuracy using a Logistic Regression model under different MCAR-style corruption levels (0% to 95%) for MNIST, F-MNIST, Flowers and CIFAR. We observe that APCs are able to generate embeddings even under high corruption levels that are still linearly separable. Neural encoders (VAE/MIWAE) on the other hand quickly collapse their embedding representation and thus drop in downstream task performance, when data is missing.

[2020]. These models were evaluated on image reconstruction tasks with varying levels of data missingness: 0%, 70%, and 90% using MCAR-style patterns, and two MAR-style patterns (missing right-half, missing center-square), as illustrated in Appendix Figure 9. Initial experiments revealed that the base NVAE model tended to either keep the zero-imputed values or exhibit failure modes such as oversaturation, not seen in previous VAE experiments. To address this, we finetuned each NVAE model (labeled "Finet." in Appendix Figure 9) on its respective dataset with MCAR patterns of $p = 50\%$ uniformly missing pixels for an additional 25% of its original training epochs. This finetuning process requires careful calibration, as finetuning degrades the model's original reconstruction capabilities or can lead to complete collapse. As depicted in Appendix Figure 9, the base NVAE achieves nearly perfect reconstructions on complete data, whereas the finetuned model exhibits significantly poorer performance. This observation is supported by a substantial increase in bits-per-dimension (BPD) on the full test sets: from 2.91 to 16.93 for CIFAR, and from 2.27 to 104.26 for CelebA, for the base to finetuned models respectively. Although the finetuned model demonstrates some ability to reconstruct images with 70% missing pixels, it remains susceptible to similar failure modes as the base model at 90% missingness and is unable to reconstruct MAR-style corruptions.

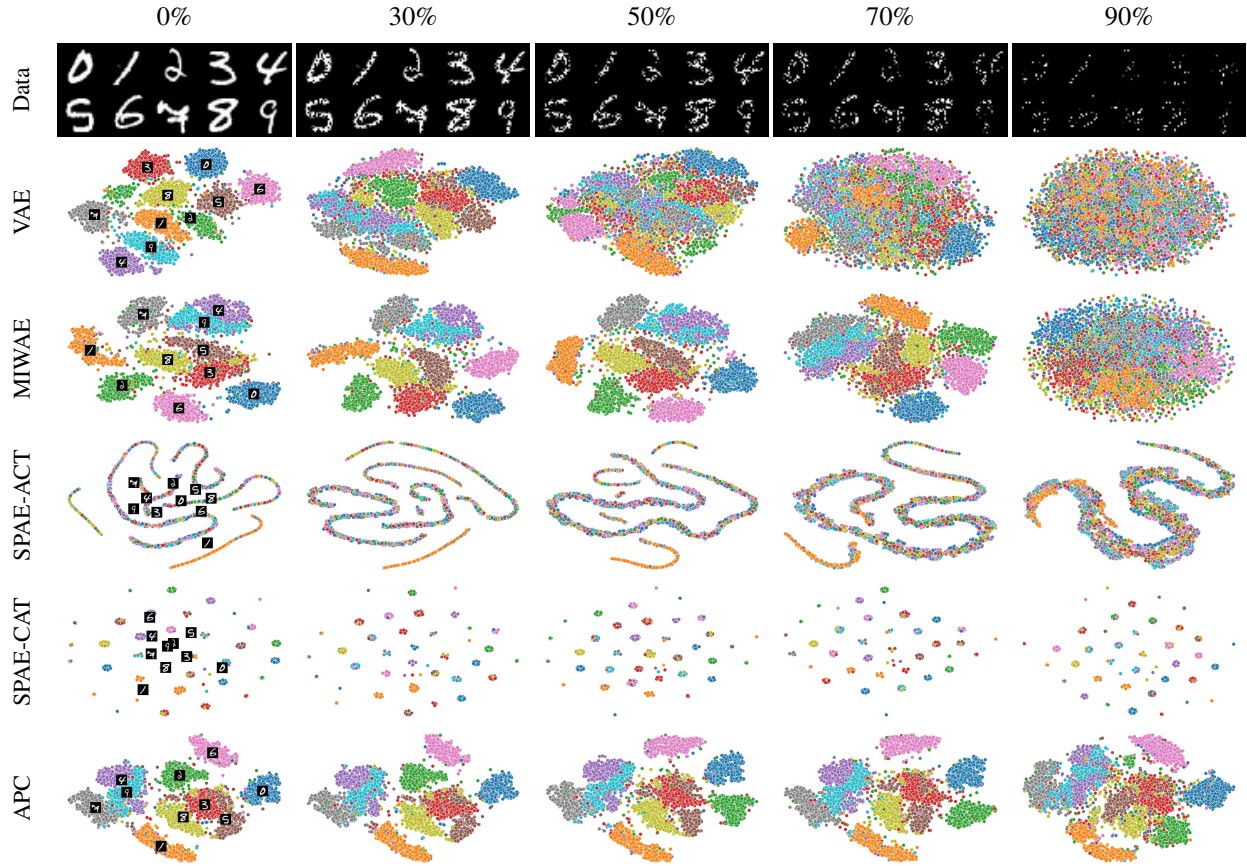

Figure 7: T-SNE projections of MNIST model embeddings for different corruption levels of missing data. Each dot is an MNIST test data point and colored according to its class. While neural encoders (*AE) initially (0%) are able to separate different classes in the embedding space, their representation starts to heavily degrade with increasing corruption levels until no separation is possible. In contrast, APCs are able to keep a stable embedding space even when corruption is high.

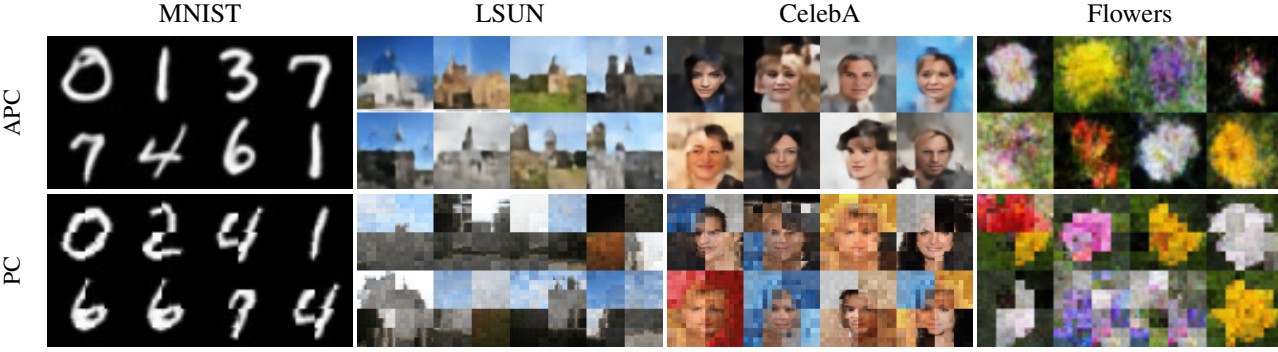

Figure 8: APC and vanilla PC samples from models trained on MNIST, LSUN (Church), CelebA, and Flowers. APCs successfully learn the data generating distribution and are able to produce novel samples. While PC samples exhibit known visual artifacts that align with their specific circuit structure, APCs are able to circumvent these with a more flexible neural decoder.

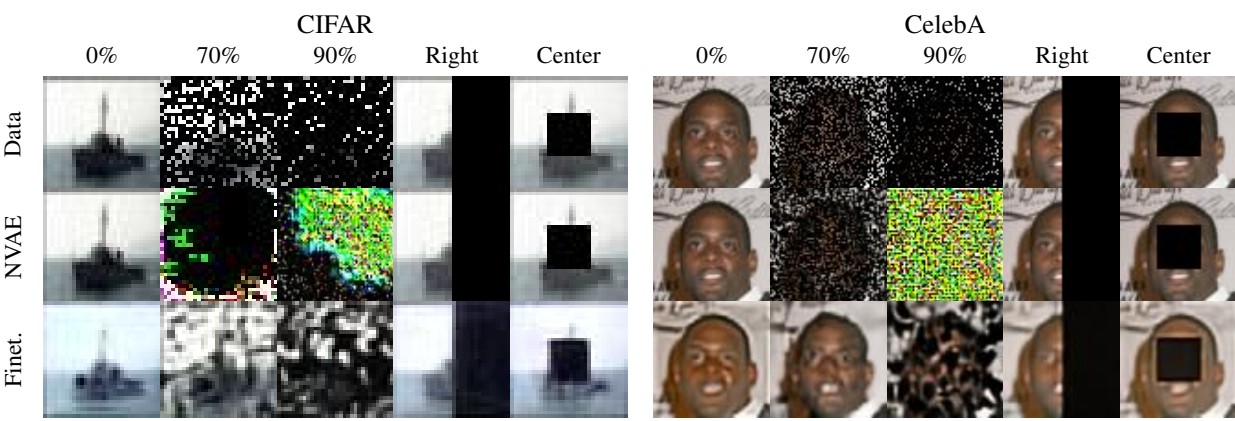

Figure 9: Exemplary NVAE and MCAR-finetuned NVAE reconstructions on CIFAR and CelebA samples with MCAR/MAR corruption. Standard NVAE retains zero-imputed values, collapsing at high corruption (e.g., 90%). MCAR-finetuning offers marginal improvement for MCAR patterns but fails to generalize to other MAR patterns (e.g., missing right/center).

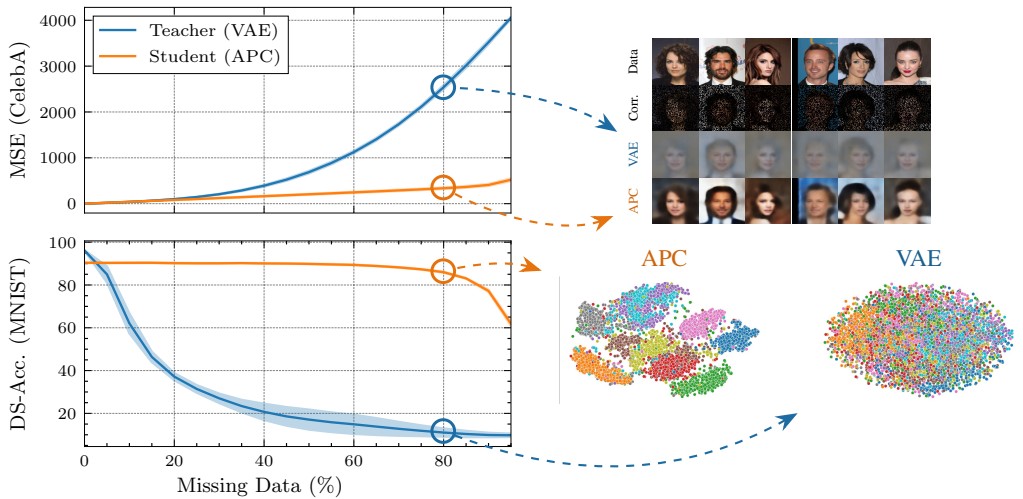

Figure 10: Data-free Knowledge Distillation from VAE to APC. The distilled APC model is able to learn the VAE's distribution and improves in reconstruction and downstream performance under data corruptions.

## B.5 APPLICATION: DATA-FREE KNOWLEDGE DISTILLATION

Pre-trained autoencoding models are increasingly shared online and offer powerful generative capabilities. However, a significant limitation, as demonstrated in our earlier evaluations (Appendices B.1 to B.3), is their often fragile performance when confronted with MCAR and MAR corrupted input data. This raises a practical question: can we transfer the generative knowledge of a pretrained VAE to our APC framework when the original training data is unavailable? Knowledge distillation (KD) has emerged as a powerful technique for model compression, adaptation, and continual learning by transferring knowledge from a teacher model to a student model [Hinton et al., 2015]. KD has seen an increasing amount of application in natural language processing [Tang et al., 2019, Jiao et al., 2020, Mou et al., 2016], computer vision [Liu et al., 2017, Zhou et al., 2018, Yim et al., 2017], and speech recognition [Hinton et al., 2015, Lu et al., 2017, Ramsay et al., 2019]. Traditionally, this process requires access to the original training data, which may not always be available due to privacy concerns, proprietary restrictions, or storage constraints. Data-free knowledge distillation seeks to overcome these challenges by extracting transferable knowledge solely from the teacher's learned representations, without requiring access to original data. Approaches range from generative adversarial network [Goodfellow et al., 2014] based and inspired methods [Chen et al., 2019, Han et al., 2021] to those that first generate synthetic data through specific distillation objectives and then train students on this synthetic data [Mordvintsev et al., 2015, Yin et al., 2020, Braun et al., 2023], align intermediate features between teacher and student [Wang, 2021a] to black-box hard-label based methods [Wang, 2021b].

In this section, we explore the capabilities of Autoencoding Probabilistic Circuits as robust data-free knowledge distillers. Given the explicit probabilistic nature and tractable inference in APCs, we investigate their potential as student models to distill knowledge from generative latent variable models, such as VAEs, while circumventing the need for the original dataset. This approach allows us to examine whether APCs can learn the data distribution captured by a VAE teacher, while additionally benefitting from APCs robustness shown in the previous sections. We outline the data-free knowledge distillation procedure in Appendix Algorithm 2. In short, we sample an embeddings from the APC prior $p_{\mathcal{C}}(\mathbf{Z})$ which is used by the teacher VAE to generate a synthetic data samples. We then train the APC according to the same objectives as outlined in Section 1 by treating the synthetic samples as reconstruction ground truth.

In Appendix Figure 10 we show the results of knowledge distillation from a VAE to an APC of similar capacity and the same decoder architecture illustratively on two example datasets. The top graph quantifies reconstruction error on CelebA, measured as the mean squared error (MSE), as a function of MCAR-style missing data percentage. The teacher VAE exhibits a sharp increase in MSE as data corruption intensifies, whereas the student APC maintains lower reconstruction error, demonstrating its effectiveness in knowledge distillation and showing the same characteristics of APCs observed in the previous sections when the APC is trained with original data supervision. In addition, we highlight the qualitative results of reconstructions with 80% randomly missing data on the right, where the APC student model more faithfully reconstructs corrupted images compared to its teacher VAE model. The bottom graph evaluates downstream task performance on MNIST,

measured by logistic regression classification accuracy using embeddings obtained from the models. As missing data increases, as observed earlier, the teacher VAE's performance rapidly deteriorates, whereas the student APC maintains high accuracy, indicating superior robustness of its learned embeddings towards corruptions. We once more highlight these results qualitatively on the right as t-SNE visualizations of the embedding spaces at 80% missing data, which further reinforces our findings: while the teacher VAE embeddings collapse into a dispersed and unstructured distribution, the student APC embeddings maintain well-separated clusters, suggesting better downstream utility. To provide further evidence, we repeat these experiments for every image and tabular dataset explored in Appendix B and show results for teacher and student reconstructions and downstream task accuracy under full evidence (Full Evi.) and average mean squared error area (MSE) over an increasing level of corruption (0% to 95% in 5% steps) in Appendix Table 8. The results confirm, that APCs closely match the VAE's performance under full evidence while surpassing it in robustness to missing data, achieving lower reconstruction errors and higher downstream accuracy across most datasets. This highlights APCs' ability to effectively distill generative knowledge from VAEs without access to original training data, while enhancing robustness to data corruption.

Table 3: Ablation study examining the contribution of key APC framework components: differentiable sampling for end-to-end training (Diff.), neural decoder (NN-Dec.), KLD embedding regularization ($\mathbf{Z}$-Reg.), and joint data-embedding log-likelihood maximization ($p(\mathbf{x}, \mathbf{z})$). Performance is evaluated using mean squared reconstruction error (MSE $\downarrow$), and downstream classification accuracy (DS-Acc. $\uparrow$) under full evidence conditions (Full Evi.), and average MSE metrics measuring robustness.

| | Diff. | NN-Dec. | E-Reg. | $p(\mathbf{x}, \mathbf{z})$ | MSE ($\downarrow$) | | DS-Acc ($\uparrow$) | |
| --- | --- | --- | --- | --- | --- | --- | --- | --- |
| | | | | | Full Evi. | MCAR | Full Evi. | MCAR |
| **MNIST** | ✓ | ✗ | ✗ | ✗ | $79.98_{\pm 10.3}$ | $63.84_{\pm 7.36}$ | $69.28_{\pm 2.69}$ | $39.16_{\pm 5.43}$ |
| | ✗ | ✓ | ✓ | ✓ | $55.21_{\pm 1.24}$ | $26.57_{\pm 0.48}$ | $59.69_{\pm 3.05}$ | $58.01_{\pm 3.11}$ |
| | ✓ | ✗ | ✓ | ✓ | $20.73_{\pm 0.66}$ | $17.12_{\pm 0.30}$ | $50.70_{\pm 3.50}$ | $45.79_{\pm 3.24}$ |
| | ✓ | ✓ | ✗ | ✓ | $8.18_{\pm 0.07}$ | $8.43_{\pm 0.02}$ | $88.13_{\pm 0.22}$ | $82.94_{\pm 0.19}$ |
| | ✓ | ✓ | ✓ | ✗ | $7.45_{\pm 0.19}$ | $19.87_{\pm 1.14}$ | $85.98_{\pm 1.77}$ | $65.58_{\pm 3.65}$ |
| | ✓ | ✓ | ✓ | ✓ | $\mathbf{4.13}_{\pm 0.10}$ | $\mathbf{5.06}_{\pm 0.05}$ | $\mathbf{88.32}_{\pm 0.14}$ | $\mathbf{86.85}_{\pm 0.11}$ |
| **CIFAR** | ✓ | ✗ | ✗ | ✗ | $441.32_{\pm 17.5}$ | $291.15_{\pm 7.43}$ | $34.76_{\pm 0.51}$ | $24.32_{\pm 0.51}$ |
| | ✗ | ✓ | ✓ | ✓ | $193.77_{\pm 0.00}$ | $92.07_{\pm 0.00}$ | $29.80_{\pm 0.00}$ | $29.51_{\pm 0.00}$ |
| | ✓ | ✗ | ✓ | ✓ | $60.55_{\pm 0.86}$ | $68.31_{\pm 0.93}$ | $33.09_{\pm 0.58}$ | $29.24_{\pm 0.64}$ |
| | ✓ | ✓ | ✗ | ✓ | $25.45_{\pm 0.20}$ | $22.52_{\pm 0.11}$ | $36.39_{\pm 0.44}$ | $31.11_{\pm 0.15}$ |
| | ✓ | ✓ | ✓ | ✗ | $35.70_{\pm 0.13}$ | $67.52_{\pm 0.56}$ | $33.71_{\pm 0.27}$ | $25.79_{\pm 0.50}$ |
| | ✓ | ✓ | ✓ | ✓ | $\mathbf{16.29}_{\pm 0.14}$ | $\mathbf{20.64}_{\pm 0.08}$ | $\mathbf{37.61}_{\pm 0.27}$ | $\mathbf{36.90}_{\pm 0.14}$ |

## B.6 ABLATION STUDIES

Following an extensive evaluation of APCs across a diverse range of datasets and against circuit-based, neural, and classic models in the previous sections, we now conduct an ablation study to isolate and quantify the contribution of each individual component within our framework, thereby strengthening the understanding of APCs. As shown in Appendix Table 3, we analyze both the removal of all components simultaneously and the exclusion of individual components while maintaining others. We evaluate four key architectural elements: (1) differentiable sampling (Diff.), which enables end-to-end gradient flow between the encoder circuit and decoder network; (2) the neural decoder (NN-Dec.), as opposed to using the circuit itself for reconstruction; (3) embedding regularization via KLD against a standard normal prior ($\mathbf{Z}$-Reg.); and (4) joint data-embedding log-likelihood maximization ($p(\mathbf{x}, \mathbf{z})$). For each configuration, Appendix Table 3 reports mean squared error (MSE) and downstream classification accuracy (DS-Acc.) under full evidence (Full Evi.) conditions, along with the MCAR-style corruptions for both MNIST and CIFAR datasets. We deliberately selected these two datasets to evaluate our components across different complexity levels: MNIST serves as a relatively simple benchmark with clear digit structures, while CIFAR represents a more challenging natural image dataset with complex objects, backgrounds, and color variations. To ensure fair comparison across all configurations, we maintain consistent model capacity throughout the experiments. Specifically, for configurations where NN-Dec. = ✗ (indicating the absence of a neural decoder), we double the encoder size to match the total parameter count of a complete APC model, where capacity is evenly distributed between encoder and decoder.

The ablation results in Appendix Table 3 highlight the indispensable role of each component in the APC framework. The baseline configuration (first row) is a circuit which performs both encoding and decoding while being optimized solely through reconstruction error minimization, mimicking a vanilla autoencoding scheme without additional objectives. Differentiable sampling (Diff.) is fundamental, as it ensures end-to-end gradient flow from the neural decoder back to the probabilistic encoder, optimizing the learned representations for reconstruction. Without this connectivity, training fails to propagate meaningful gradients, resulting in catastrophic degradation – MSE increases by factors of $13.4\times$ on MNIST and $11.9\times$ on CIFAR. Similarly, the neural decoder (NN-Dec.) plays a pivotal role in reconstruction and downstream performance, as its absence leads to severe drops in reconstruction quality, increasing MSE by $5\times$ and $3.7\times$ on MNIST and CIFAR, respectively. This impact is even more pronounced in corruption MCAR metrics, demonstrating the decoder's importance for robustness against missing data. Embedding regularization ($\mathbf{Z}$-Reg.) further refines representation learning, nearly halving the MSE on MNIST (from 8.18 to 4.13) and significantly improving the corruption MCAR MSE from 8.43 to 5.06. These results confirm that each component makes a distinct and complementary contribution to the APC framework,

with their combination yielding better performance than any partial implementation. For qualitative reconstruction examples, refer to Appendix Figure 5.

# C  ENCODING ALGORITHM

---

**Algorithm 1** APC Encoding Procedure

---

**Require:** Probabilistic circuit $\mathcal{C}$ over data $\mathbf{X}$ and embeddings $\mathbf{Z}$, an input data point $\mathbf{x} \in \mathbf{X}$

1:
2: **procedure** ENCODE($\mathcal{C}$: circuit, $\mathbf{x}$: data)
3:     FORWARD(ROOT($\mathcal{C}$), $\mathbf{x}$)                                  ▷ Forward pass; at each sum unit we store its inputs
4:     $\big[\mathbf{x}, \mathbf{z}\big] = $ SAMPLE(ROOT($\mathcal{C}$))                          ▷ Sample from conditioned PC $\mathcal{C}|_{\mathbf{X}=\mathbf{x}}$
5:     **return** $\mathbf{z}$

6:
7: **procedure** FORWARD($n$: unit, $\mathbf{x}$: data)                              ▷ General forward pass for circuit units
8:     **if** $n == \bigwedge$ **then**
9:         **if** $\mathbf{x}_n$ is missing **then**
10:             **return** $1.0$                                          ▷ Marginalize missing inputs
11:         **else**
12:             **return** $p_n(\mathbf{x}_n)$                                 ▷ Evaluate PDF of input units
13:     $\boldsymbol{\gamma} \leftarrow \big[c(\mathbf{x}_c)$ for $c \in$ IN$(n)\big]$                       ▷ Collect inputs for sum/product units
14:     $n.\boldsymbol{\gamma} \leftarrow \boldsymbol{\gamma}$                                      ▷ Cache inputs for conditional sampling pass
15:     **if** $n == \bigotimes$ **then**
16:         **return** $\prod_{c \in \text{IN}(n)} \gamma_c$                                    ▷ Product unit
17:     **if** $n == \bigoplus$ **then**
18:         **return** $\sum_{c \in \text{IN}(n)} \gamma_c \cdot \theta_n^c$                                 ▷ Sum unit

19:
20: **procedure** SAMPLE($n$: unit)                                       ▷ Sampling pass for circuit units
21:     **if** $n == \bigwedge$ **then**
22:         **return** $\mathbf{x}_n \sim p_n(\mathbf{X}_n)$                       ▷ Sample from PDF represented by input unit
23:     **if** $n == \bigotimes$ **then**
24:         **return** CONCAT($\big[$SAMPLE($c$) for $c \in$ IN$(n)\big]$)             ▷ Product unit simply sample all inputs
25:     **if** $n == \bigoplus$ **then**
26:         $\boldsymbol{\theta}_n' \leftarrow$ CONDITION_WEIGHTS($\boldsymbol{\theta}_n, n.\boldsymbol{\gamma}$)          ▷ Condition weights on forward pass likelihoods
27:         $\mathbf{s} \leftarrow$ SIMPLE($\boldsymbol{\theta}_n'$)                           ▷ Sample one-hot encoded index
28:         **return** DOT($\mathbf{s}, \big[$SAMPLE($c$) for $c \in$ IN$(n)\big]$)               ▷ Dot product indexes input samples

29:
30: **procedure** CONDITION_WEIGHTS($\boldsymbol{\theta}$: weights, $\boldsymbol{\gamma}$: inputs)
31:     **for** $i \in 1 \ldots$ SIZE($\boldsymbol{\theta}$) **do**
32:         $\theta_i \leftarrow \theta_c \cdot \gamma_i$                                   ▷ Reweight weights based on input likelihoods
33:     $s = \sum_i \theta_i$                                           ▷ Compute normalization constant
34:     **for** $i \in 1 \ldots$ SIZE($\boldsymbol{\theta}$) **do**
35:         $\theta_i \leftarrow \theta_i / s$                                      ▷ Normalize weight distribution
36:     **return** $\boldsymbol{\theta}$

---

Conditional sampling in PCs, which forms the basis of our encoding procedure (Appendix Algorithm 1), consists of two passes through the circuit. First, a forward pass computes the marginal likelihood of the evidence, $p_{\mathcal{C}}(\mathbf{x})$, for a given data sample $\mathbf{x}$. During this pass, likelihoods $\boldsymbol{\gamma}$ are cached at the inputs of each sum unit. Subsequently, a sampling pass traverses the circuit from the root to the input units. At each sum unit, it uses the cached likelihoods $\boldsymbol{\gamma}$ from the forward pass to reweight the mixture parameters, effectively applying Bayes' rule to form a posterior over its inputs. It then samples a path according to these conditioned weights using the differentiable SIMPLE estimator. This process yields a sample from the conditional distribution $p_{\mathcal{C}}(\mathbf{Z} \,|\, \mathbf{X} = \mathbf{x})$, which serves as the embedding $\mathbf{z}$.

# D   EXPERIMENT AND EVALUATION PROTOCOL: ADDITIONAL DETAILS

All experiments and models are implemented in PyTorch [Ansel et al., 2024] and PyTorch Lightning [Falcon and The PyTorch Lightning team, 2019]. Our implementation is available as open-source software at `https://github.com/placeholder-url`.

**Datasets**   We evaluate our models various image and tabular datasets. For image data, we include MNIST [LeCun et al., 1998], Fashion MNIST (F-MNIST) [Xiao et al., 2017], CIFAR-10 [Krizhevsky, 2009], CelebA [Liu et al., 2015], SVHN [Netzer et al., 2011], Flowers [Gurnani et al., 2017], LSUN (Church) [Yu et al., 2015], and Tiny-ImageNet [Deng et al., 2009]. Note that Tiny-ImageNet is occasionally abbreviated as ImageNet in our tables for brevity. For tabular data, we utilize the 20 datasets from the binary density estimation benchmark DEBD [Lowd and Davis, 2010, Haaren and Davis, 2012, Bekker et al., 2015, Larochelle and Murray, 2011].

**Models**   In our empirical evaluation, we compare the proposed APCs against several models to demonstrate their effectiveness. These include SPAE-ACT and SPAE-CAT [Vergari et al., 2018] as a circuit-based comparison, as well as vanilla variational autoencoder (VAE) and, to also include more missing-data specific methods MIWAE [Mattei and Frellsen, 2019] and missForest [Stekhoven and Bühlmann, 2011]. For all autoencoding models, we use an embedding dimension $d$ of 64 for MNIST and F-MNIST, 256 for all other image-based datasets, and 4,8,16,32, and 64 for tabular datasets, depending on their number of features to ensure $d \ll |\mathbf{X}|$. All models are trained under the same conditions with common and well-established practices.

**Metrics**   We report mean squared error (MSE ↓) to measure reconstruction fidelity and structured similarity index measure (SSIM ↑) to assess perceptual quality based on structural and visual similarity. To assess encoding robustness, we analyze MSE and SSIM across varying levels of MCAR-style input data corruption, where uniformly random corruption is incrementally increased from 0% to 95% in steps of 5%. We then compute the average reconstruction error over all corruption levels for each metric. To ensure that results are not specific to MCAR-style missing data, Appendix Table 4 presents evaluations using nine additional MAR-style corruption variants.

**Missing Data**   For vanilla VAEs, missing values are imputed using a constant zero value. We also explored alternative imputation strategies, including mean imputation and learned normalization (similar to LayerNorm). In our experiments on MNIST, mean imputation offered improved reconstruction quality at the cost of a reduction in downstream task accuracy. For more challenging datasets like SVHN and CIFAR, the choice of imputation method did not yield statistically significant differences in performance. All other methods handle missing data without the need for imputation.

**Model Capacity**   For all experiments, we ensure that the model encoder and decoder networks are approximately the same size and depth across all models. We use the same neural decoder architecture across all autoencoding models to ensure that performance differences can be solely attributed to the encoder.

**Circuit Input Units**   For circuit-based encoders, we use model data distributions on images with Binomial input units, and Bernoulli input units for the DEBD tabular datasets. Embedding distributions are chosen as Gaussian input units in all cases. We want to highlight once more that the APC framework, in principle, allows for the choice of arbitrary data and *embedding* distributions.

**Sum-Product Autoencoding**   For SPAE, we retrieve the correctly sized embedding vector from a layer in the circuit graph, which consists of exactly $d$ units (embedding size). In addition, when evaluating reconstructions, we report results only for SPAE (instead of SPAE-ACT and SPAE-CAT) because the reconstruction output is identical for both SPAE variants, regardless of whether activation embeddings (ACT) or categorical embeddings (CAT) are used. This equivalence is formally established in Proposition 3 of Vergari et al. [2018]. However, in all other evaluations, we distinguish between SPAE-ACT and SPAE-CAT, since their embeddings are inherently different when used in e.g., downstream tasks.

**Autoencoder Training**   Each model is trained for 10,000 iterations using the AdamW optimizer [Kingma and Ba, 2015, Loshchilov and Hutter, 2017], and convergence was confirmed for all models by the end of this training period. We use the mean squared error (MSE) as $\mathcal{L}_{\text{REC}}$ for all models. Training is carried out with a batch size of 512, except for CelebA where a batch size of 256 is used due to larger model sizes and VRAM constraints. The initial learning rate is set to 0.1 for APCs and 0.005 for AEs and VAEs. The learning rate is reduced by a factor of 10 at 66% and 90% of the training progress. Additionally, to enhance stability and avoid numerical issues or exploding gradients during the training phase, we

utilize an exponential learning rate warmup over the first 2% of training iterations. Empirically, this approach mitigates random training difficulties without affecting the final performance across different random seeds. While, for APCs, we could potentially pretrain the PC-encoder with MLE on the marginal $p(\mathbf{x})$ to speed up convergence, we refrain from doing so in our experiments, to keep the comparison between APC and VAE-based models as fair as possible. All models, with the exception of MIWAE, were trained on the complete datasets without missing values. For MIWAE, training was performed using MCAR-style data in which 50% of entries were missing.

**Downstream Task Training**  We train the logistic regression downstream task models with a batch size of 512 for 5,000 iterations on dataset embeddings from the respective encoders. We employed an initial learning rate of 0.05, which was reduced by a factor of 0.1 at 66% and 90% of the training progress, along with an exponential learning rate warmup over the first 2% of iterations. For SPAE-ACT models specifically, we found it necessary to use the AdamW optimizer with a learning rate of 0.01 for 100,0000 iterations (20× longer) and normalize the embeddings to achieve comparable downstream task performance better than a random guessing baseline.

## D.1   MODEL ARCHITECTURES

**Neural Encoder and Decoder.**  For tabular data, we employ a simple linear feed-forward style network with four hidden layers and leaky ReLU ($\alpha = 0.1$) activations as encoder and decoder. We use convolutional and residual layers with ReLU activations instead for image data. For further details, we refer the reader to `APC.models.{encoder,decoder}.nn_{encoder,decoder}.py` files in our source code repository.

**EinsumNetwork Encoder for Tabular Data.**  We use EinsumNetworks [Peharz et al., 2020] as circuit structure for APCs and SPAEs with Bernoulli input units for the tabular DEBD dataset [Lowd and Davis, 2010, Haaren and Davis, 2012, Bekker et al., 2015, Larochelle and Murray, 2011]. For APCs, embedding input units are inserted at the lowest layer and randomly shuffled with the data distribution input units. For all experiments, we keep a depth of 4, with a single repetition and 32 input unit distributions, and the sum units per scope at each layer.

**ConvPc Encoder for Image Data.**  For image data, we construct a simple "convolutional" circuit for APCs and SPAEs, where we first map each pixel to its density represented by an input unit layer with 256 output channels (number of input unit distributions per scope). We then successively reduce the height and width by a factor of 2 with product layers, which builds the product over all scopes in disjoint neighboring windows (think of non-overlapping convolution windows with stride being the window size). After each product layer, a sum layer maps all channels of each scope to a vector of sum unit outputs (think of convolution in- and out-channels). We repeat this until we end up with a single dimension where the scopes cover the full image input. This allows us to choose a certain depth (how often do we want to repeat the product-sum combination), and the choice of number of sum output units per sum layer. Embedding input units are inserted randomly at the lowest layer with product units attached to $|\mathbf{Z}|$ data input units, constructing $|\mathbf{Z}|$ combinations of $p_{\mathcal{C}}(x_i)\,p_{\mathcal{C}}(z_j)$. More advanced strategies for embedding insertions remain an avenue for future investigation; for example, embedding input units could be decoupled and integrated at varying hierarchical levels within the circuit.

# E   RECONSTRUCTION PERFORMANCE: ADDITIONAL RESULTS

This section provides a more detailed exposition of the reconstruction capabilities of APCs and the compared models, supplementing the empirical evaluations presented in the main body. We include additional quantitative results, such as the Structured Similarity Index Measure (SSIM) for image datasets, further explore the impact of various Missing at Random (MAR) corruption patterns beyond the Missing Completely At Random (MCAR) scenarios discussed earlier, and present extended qualitative results through more comprehensive sets of reconstruction visualizations across all datasets and corruption types. Furthermore, we offer an initial exploration into the potential of APC embeddings for out-of-distribution detection by analyzing embedding likelihoods.

## E.1   MISSING AT RANDOM (MAR) STYLE CORRUPTIONS

Table 4: Model ranking per corruption type, aggregated across datasets. Each row represents a different MAR-style corruption applied to the input images. The severity of each corruption is linearly increased and the average reconstruction MSE/SSIM is measured. The models are ranked based on their robustness to these corruptions aggregated across all image datasets. The reported values represent the ranking based on the average MSE/SSIM. See Appendix Figures 14 and 15 for reconstruction visualizations.

| | MSE | | | | SSIM | | | |
|---|---|---|---|---|---|---|---|---|
| Corruption | APC | SPAE | VAE | MIWAE | APC | SPAE | VAE | MIWAE |
| left-to-right | **1** | 4 | 2 | 3 | **1** | 3 | 2 | 4 |
| right-to-left | **1** | 4 | 2 | 3 | **1** | 2 | 3 | 4 |
| top-to-bottom | **1** | 4 | 2 | 3 | **1** | 3 | 2 | 4 |
| bottom-to-top | **1** | 4 | 2 | 3 | **1** | 3 | 2 | 4 |
| border-to-center | **1** | 3 | 2 | 4 | **1** | 2 | 3 | 4 |
| center-to-border | **1** | 3 | 4 | 2 | **1** | 2 | 3 | 4 |
| horizontal-band | **1** | 3 | 4 | 2 | **1** | 4 | 2 | 3 |
| vertical-band | **1** | 2 | 3 | 4 | **1** | 3 | 2 | 4 |
| salt-and-pepper | **1** | 3 | 2 | 4 | **1** | 4 | 2 | 3 |

To further investigate model robustness beyond *randomly* missing data, Appendix Table 4 presents model rankings across different MAR-style corruption types, aggregated over all datasets. For each corruption type, we repeat the experiments presented in Appendix Table 1 for all image datasets, assign each method a rank from best (1.) to worst (4.) and then report their final rank, sorted by the average rank over all datasets. Note that we only compare autoencoding methods, as they reconstruct the full image from a low-dimensional latent space. In contrast, missForest operates directly in the input space, making it an inappropriate comparison for a ranking in this context. APCs consistently achieve the top rank for all but one corruption type and metric, indicating that their superior performance extends beyond handling missing data to various other forms of data corruption. Vanilla VAEs appear to outrank its missing-data specific variant MIWAE in both MSE nad SSIM measurements. This could be attributed to the fact, that MIWAE specifically trains on a certain type of missing data, MCAR in our case. This leads to a decreased performance in reconstruction, when the pattern of missing data substantially changes. We can visually confirm this is Appendix Figures 13 and 15, where MIWAE is better at reconstructing MCAR-style missing data than VAEs, but fails similarly when confronted with MAR-style corruptions (horizontal, vertical, center, border).

## E.2   STRUCTURED SIMILARITY INDEX (SSIM) EVALUATIONS

While Mean Squared Error (MSE) provides a quantitative measure of pixel-wise differences, the Structural Similarity Index Measure (SSIM) is also employed to offer a complementary perspective on reconstruction quality. SSIM is designed to better align with human visual perception by evaluating similarities in luminance, contrast, and structure between the original and reconstructed images. Consistent with the MSE results, the findings presented in Appendix Table 5 (complementary to Appendix Table 1) and Appendix Figure 11 (complementary to Fig. 3) demonstrate that APCs significantly outperform all other compared methods in preserving image fidelity across all datasets, even as the proportion of missing data increases.

Table 5: Reconstruction performance on eight image datasets under increasing percentages of MCAR-style randomly missing pixels (0%-95%). Results are reported as the average structured similarity index measure (SSIM, ↑) over the different corruption levels. APCs consistently outperform all other methods in maintaining reconstruction quality as the proportion of missing data increases.

|  | APC | SPAE | VAE | MIWAE | missForest |
|---|---|---|---|---|---|
| MNIST | **93.33** $\pm 0.04$ | 69.87 $\pm 0.36$ | 50.74 $\pm 0.82$ | 86.81 $\pm 1.14$ | 62.29 $\pm 0.46$ |
| F-MNIST | **87.40** $\pm 0.05$ | 68.89 $\pm 0.23$ | 51.75 $\pm 0.38$ | 78.24 $\pm 0.43$ | 54.00 $\pm 0.04$ |
| CIFAR | **78.20** $\pm 0.07$ | 50.61 $\pm 0.88$ | 61.64 $\pm 0.14$ | 74.25 $\pm 0.15$ | 57.98 $\pm 0.13$ |
| CelebA | **79.52** $\pm 0.04$ | 49.02 $\pm 0.17$ | 50.30 $\pm 0.30$ | 61.26 $\pm 0.66$ | 37.16 $\pm 0.10$ |
| SVHN | **90.29** $\pm 0.03$ | 54.88 $\pm 0.91$ | 61.79 $\pm 0.18$ | 76.23 $\pm 0.47$ | 56.86 $\pm 0.17$ |
| Flowers | **60.40** $\pm 0.16$ | 44.01 $\pm 0.82$ | 50.63 $\pm 0.43$ | 56.68 $\pm 0.11$ | 56.94 $\pm 0.10$ |
| LSUN | **77.14** $\pm 0.07$ | 50.73 $\pm 0.52$ | 55.63 $\pm 0.28$ | 59.80 $\pm 0.16$ | 27.32 $\pm 0.02$ |
| ImageNet | **71.55** $\pm 0.04$ | 50.62 $\pm 0.36$ | 47.76 $\pm 0.23$ | 61.58 $\pm 0.15$ | 50.98 $\pm 0.05$ |

## E.3 ADDITIONAL RECONSTRUCTION VISUALIZATIONS

To provide further visual insights and extend the quantitative reconstructions presented in Appendix Figure 5, we additionally show MCAR-style corruptions for 0%, 30%, 50%, 70%, and 90% uniformly random missing data in Appendix Figures 12 and 14 and horizontal bands, vertical band, center and border MAR-style corruptions in Appendix Figures 13 and 15 on all image datasets for all methods. These results further provide evidence, that APCs outperform the alternative methods in all investigated corruption scenarios, providing better reconstructions across the bench.

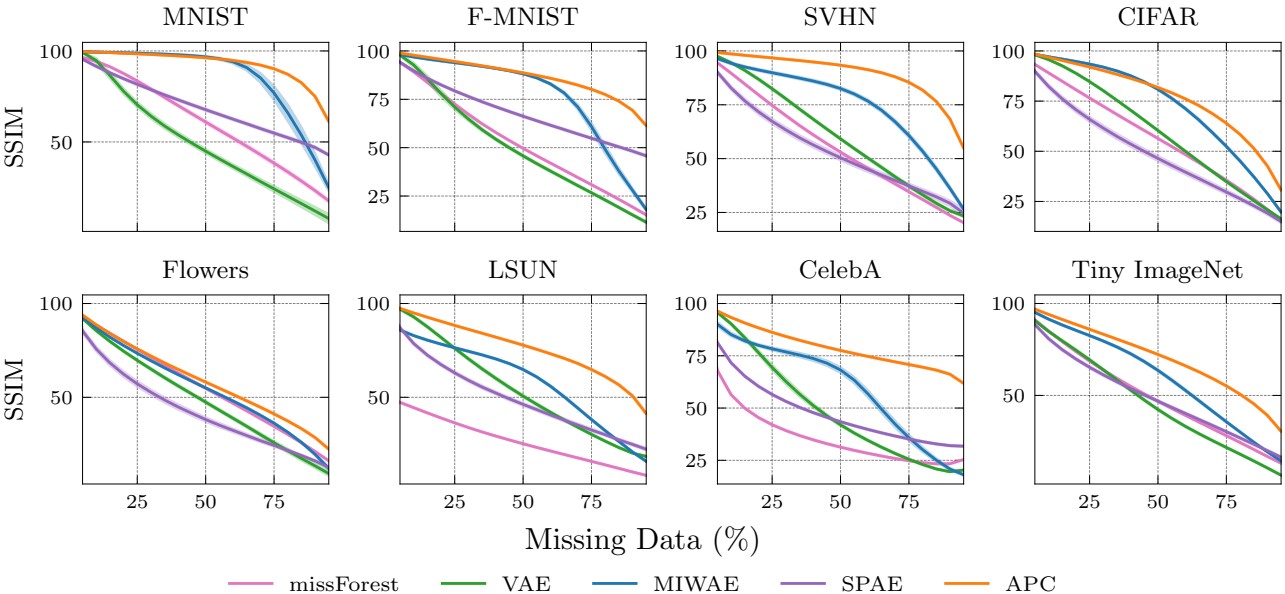

Figure 11: Reconstruction SSIM on all image datasets for increasing degrees of MCAR-style randomly missing data (0% - 95%). APCs are able to outperform all other models by large margins on all datasets with increasing fractions of missing data.

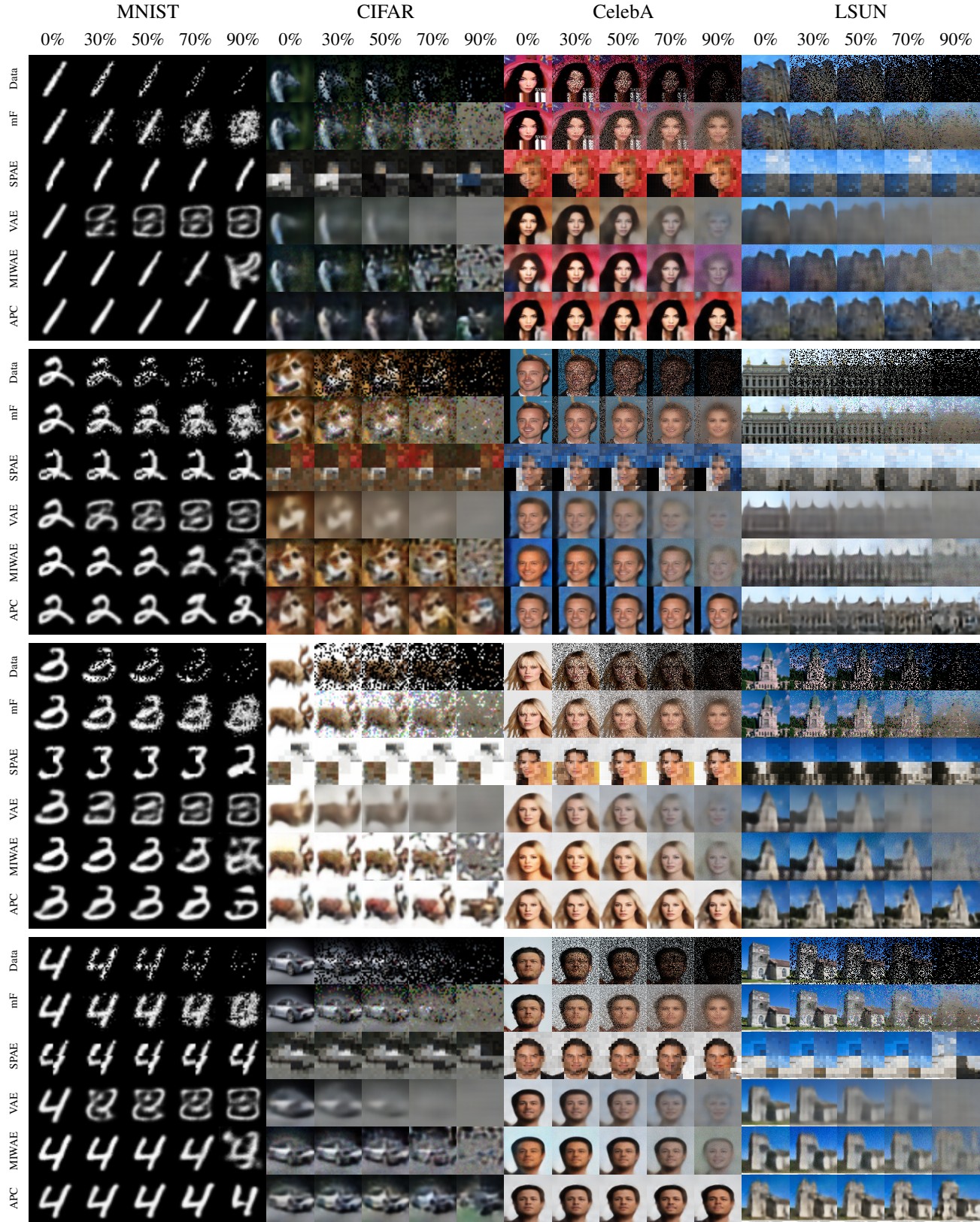

Figure 12: Reconstructions under different corruption levels for MNIST, CIFAR, CelebA and LSUN.

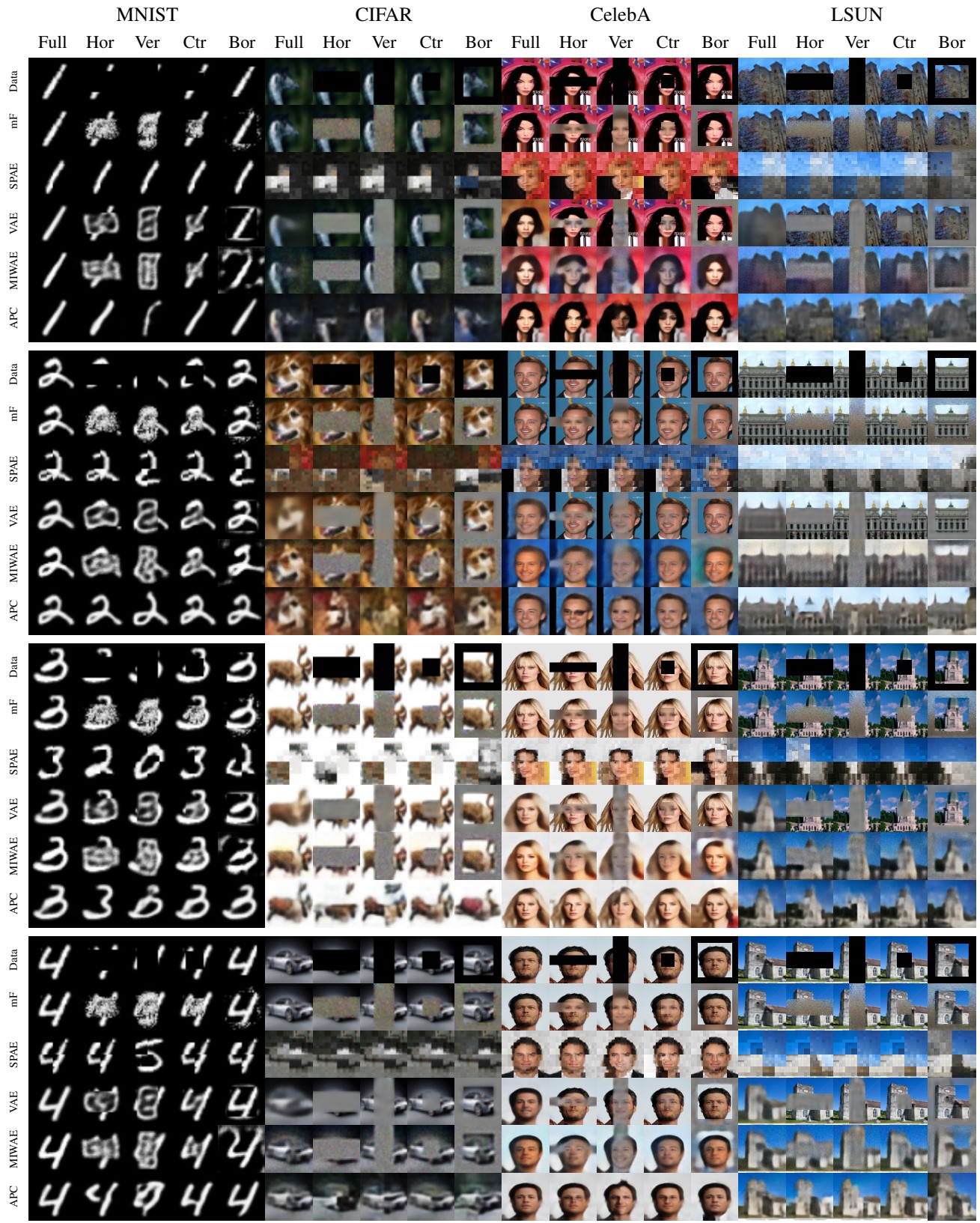

Figure 13: Reconstructions of various MAR cooruptions for MNIST, CIFAR, CelebA and LSUN.

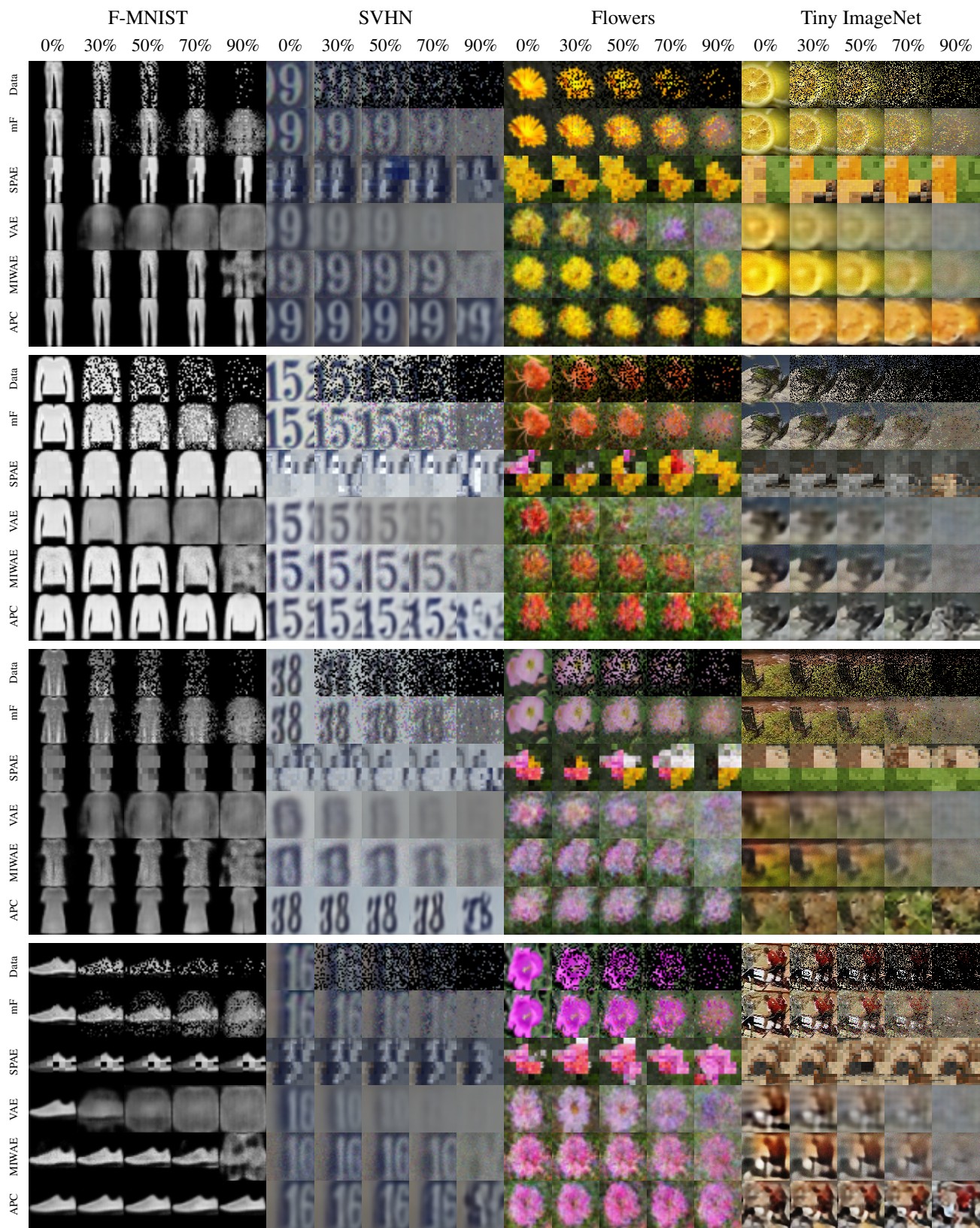

Figure 14: Reconstructions under different corruption levels for F-MNIST, SVHN, Flowers and Tiny-ImageNet.

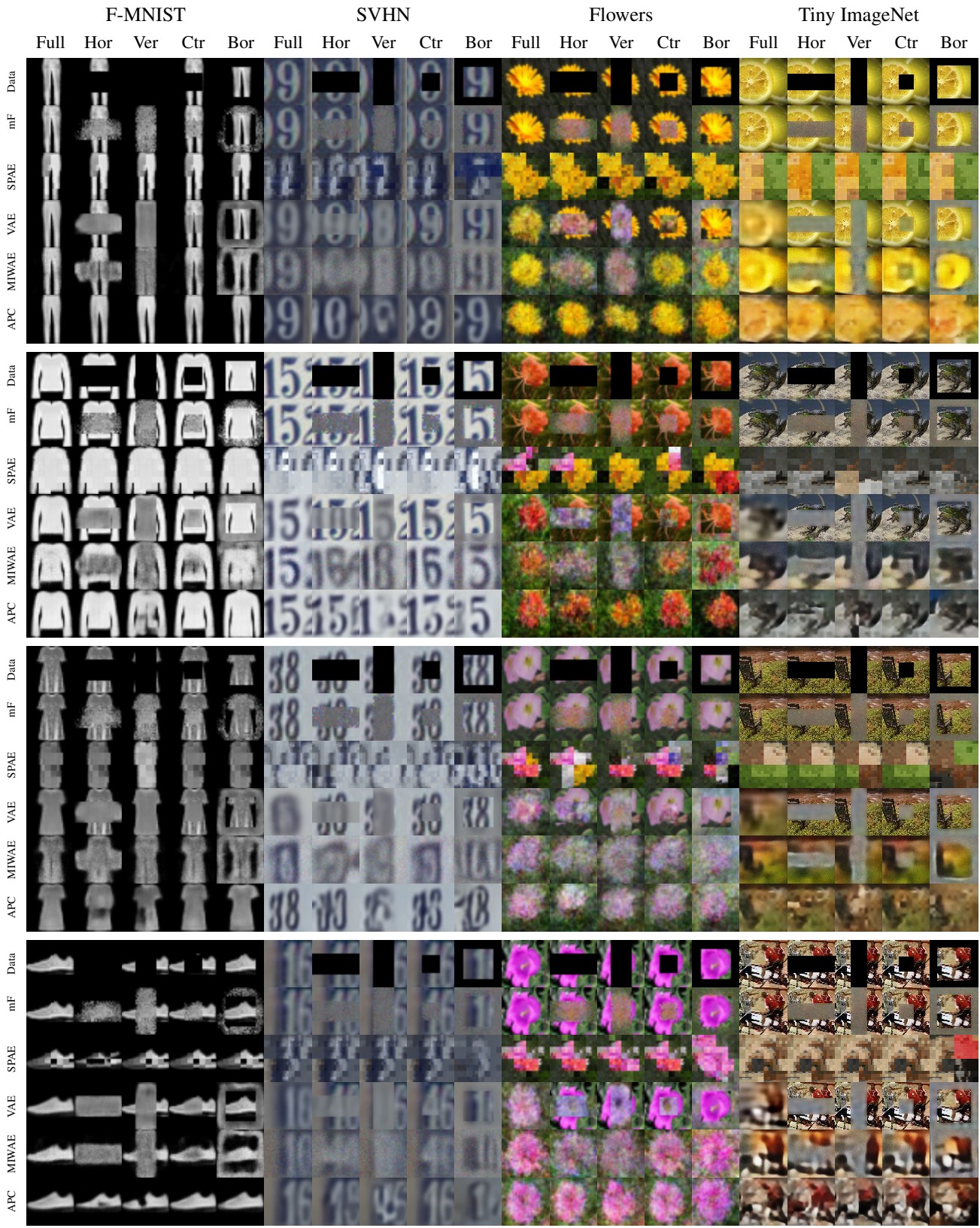

Figure 15: Reconstructions of various MAR corruptions for F-MNIST, SVHN, Flowers and Tiny-ImageNet.

Table 6: Reconstruction performance on eight image datasets under increasing percentages of randomly missing pixels (0%-95%). Results are reported as the average mean squared error (MSE, ↓) and Structural Similarity Index Measure (SSIM, ↑) for MCAR-style corruptions. APCs consistently outperform other methods in maintaining reconstruction quality as the proportion of missing data increases.

| | APC | $APC_{pc}$ | SPAE | $APC\text{-}SPAE_{pc}$ | $APC\text{-}SPAE\text{-}ACT_{nn}$ | $APC\text{-}SPAE\text{-}CAT_{nn}$ |
|---|---|---|---|---|---|---|
| | | | MSE ↓ | | | |
| MNIST | $\mathbf{5.07}_{\pm 0.04}$ | $17.17_{\pm 0.41}$ | $28.46_{\pm 0.43}$ | $49.82_{\pm 3.01}$ | $14.71_{\pm 0.25}$ | $20.07_{\pm 1.00}$ |
| F-MNIST | $\mathbf{4.33}_{\pm 0.02}$ | $17.09_{\pm 0.89}$ | $18.96_{\pm 0.11}$ | $49.46_{\pm 2.75}$ | $12.79_{\pm 0.62}$ | $17.97_{\pm 3.58}$ |
| CIFAR | $\mathbf{20.65}_{\pm 0.09}$ | $59.22_{\pm 0.30}$ | $78.32_{\pm 3.09}$ | $102.74_{\pm 0.93}$ | $31.88_{\pm 0.53}$ | $42.44_{\pm 0.26}$ |
| CelebA | $\mathbf{166.89}_{\pm 0.65}$ | $644.36_{\pm 14.2}$ | $1318.73_{\pm 6.37}$ | $2969.30_{\pm 235.}$ | $1173.57_{\pm 156.}$ | $1220.76_{\pm 318.}$ |
| SVHN | $\mathbf{4.87}_{\pm 0.00}$ | $32.27_{\pm 0.38}$ | $35.38_{\pm 1.92}$ | $78.01_{\pm 4.37}$ | $17.59_{\pm 0.24}$ | $18.74_{\pm 0.35}$ |
| Flowers | $\mathbf{45.22}_{\pm 0.14}$ | $58.65_{\pm 0.72}$ | $119.31_{\pm 5.23}$ | $110.05_{\pm 6.19}$ | $72.46_{\pm 0.60}$ | $75.05_{\pm 2.91}$ |
| LSUN | $\mathbf{63.70}_{\pm 0.20}$ | $185.92_{\pm 3.62}$ | $322.22_{\pm 3.48}$ | $489.87_{\pm 18.7}$ | $194.10_{\pm 16.6}$ | $185.69_{\pm 3.11}$ |
| ImageNet | $\mathbf{118.70}_{\pm 0.16}$ | $276.92_{\pm 3.07}$ | $365.06_{\pm 2.20}$ | $566.89_{\pm 17.2}$ | $369.20_{\pm 35.7}$ | $267.96_{\pm 20.6}$ |
| | | | SSIM ↑ | | | |
| MNIST | $\mathbf{93.33}_{\pm 0.04}$ | $79.87_{\pm 0.57}$ | $69.87_{\pm 0.36}$ | $45.21_{\pm 3.15}$ | $82.63_{\pm 0.35}$ | $64.52_{\pm 2.46}$ |
| F-MNIST | $\mathbf{87.40}_{\pm 0.05}$ | $69.44_{\pm 1.00}$ | $68.89_{\pm 0.23}$ | $39.11_{\pm 1.18}$ | $75.81_{\pm 0.58}$ | $60.89_{\pm 4.83}$ |
| CIFAR | $\mathbf{78.20}_{\pm 0.07}$ | $58.23_{\pm 0.31}$ | $50.61_{\pm 0.88}$ | $44.18_{\pm 0.17}$ | $71.41_{\pm 0.18}$ | $61.31_{\pm 0.14}$ |
| CelebA | $\mathbf{79.52}_{\pm 0.04}$ | $58.79_{\pm 0.33}$ | $49.02_{\pm 0.17}$ | $26.71_{\pm 1.17}$ | $49.03_{\pm 0.83}$ | $44.56_{\pm 3.77}$ |
| SVHN | $\mathbf{90.29}_{\pm 0.03}$ | $62.82_{\pm 0.24}$ | $54.88_{\pm 0.91}$ | $39.22_{\pm 0.76}$ | $75.60_{\pm 0.25}$ | $67.86_{\pm 0.36}$ |
| Flowers | $\mathbf{60.40}_{\pm 0.16}$ | $59.16_{\pm 0.15}$ | $44.01_{\pm 0.82}$ | $40.80_{\pm 0.87}$ | $50.35_{\pm 0.18}$ | $51.86_{\pm 1.12}$ |
| LSUN | $\mathbf{77.14}_{\pm 0.07}$ | $59.09_{\pm 0.38}$ | $50.73_{\pm 0.52}$ | $37.45_{\pm 0.93}$ | $58.72_{\pm 0.89}$ | $55.39_{\pm 0.39}$ |
| ImageNet | $\mathbf{71.55}_{\pm 0.04}$ | $55.68_{\pm 0.29}$ | $50.62_{\pm 0.36}$ | $37.90_{\pm 0.77}$ | $55.43_{\pm 0.45}$ | $53.50_{\pm 0.97}$ |

# F   AUTOENCODING PROBABILISTIC CIRCUIT VARIANTS

Having established the contribution of each component through our ablation studies in the previous section, we now build upon the core APC framework and additionally explore several alternative configurations motivated by previous work of Sum-Product Autoencoding [Vergari et al., 2018]. These variations allow us to examine the impact of different encoding and decoding strategies within the APC framework. Specifically, we consider the following:

- **APC $_{pc}$**: We replace the neural network decoder with the circuit encoder $\mathcal{C}$ itself. Leveraging the flexibility of PCs, reconstructions $\hat{x}$ are generated by sampling from the conditional distribution $p_{\mathcal{C}}(\mathbf{X} \mid \mathbf{Z})$. Differentiable sampling enables end-to-end training of this configuration, using the same objective as standard APCs to optimize the circuit for both encoding and decoding.

- **APC-SPAE$_{pc}$**: This configuration trains an APC using the SPAE scheme for encoding and decoding, employing the circuit $\mathcal{C}$ for both. This means the encoding and decoding processes are based on ACT embeddings (CAT embeddings lead to identical reconstructions as per Vergari et al. [2018]). Note, that KLD regularization of embeddings is omitted due to the different nature of ACT and CAT embeddings compared to the default APC formulation, and $\mathcal{L}_{\text{NLL}}$ is limited to $p_{\mathcal{C}}(\mathbf{X})$, as explicit embedding representations $\mathbf{Z}$ are not directly modeled in the circuit in this variant.

- **APC-SPAE-ACT$_{nn}$**: We combine the ACT encoding from SPAE with a neural decoder. Similar to APC-SPAE$_{pc}$, KLD regularization is not applied and $\mathcal{L}_{\text{NLL}}$ is limited to $p_{\mathcal{C}}(\mathbf{X})$. In our experiments we found that due to circuit activation statistics, it was necessary to add a batch normalization layer on top of the circuit activations for the training with a neural decoder to be stable.

- **APC-SPAE-CAT$_{nn}$**: Analogous to APC-SPAE-ACT$_{nn}$, this variant utilizes the CAT embedding encoding approach from SPAE, coupled with a neural decoder for reconstruction. Training follows the same strategy as APC-SPAE-ACT$_{nn}$, excluding KLD regularization and limiting $\mathcal{L}_{\text{NLL}}$ to $p_{\mathcal{C}}(\mathbf{X})$.

In Appendix Tables 6 and 7 we report the reconstruction performance of all APC variants on image and tabular datasets with progressively increasing proportions of randomly missing pixels, contrasting them with full APC and SPAE models. We evaluate reconstruction quality using the average MCAR-style corruption reconstruction error for both mean squared error (MSE) and the Structural Similarity Index Measure (SSIM), which capture the impact of varying corruption levels. As shown,

Table 7: Reconstruction performance on the 20 DEBD binary tabular datasets under increasing percentages of randomly missing data (0%-95%). Results are reported as the average reconstruction error of MCAR-style corruptions for mean squared error (MSE, ↓). APCs achieve the best reconstruction performance on 19 of the 20 datasets.

| | APC | $APC_{pc}$ | SPAE | $APC\text{-}SPAE_{pc}$ | $APC\text{-}SPAE\text{-}ACT_{nn}$ | $APC\text{-}SPAE\text{-}CAT_{nn}$ |
|---|---|---|---|---|---|---|
| accidents | $\mathbf{6.16}_{\pm 0.04}$ | $29.16_{\pm 0.52}$ | $12.55_{\pm 0.86}$ | $22.16_{\pm 1.95}$ | $8.31_{\pm 0.31}$ | $6.75_{\pm 0.01}$ |
| ad | $\mathbf{5.57}_{\pm 0.03}$ | $346.14_{\pm 5.94}$ | $275.00_{\pm 68.1}$ | $374.18_{\pm 15.7}$ | $7.88_{\pm 0.41}$ | $\mathbf{5.57}_{\pm 0.03}$ |
| baudio | $\mathbf{6.50}_{\pm 0.03}$ | $23.31_{\pm 1.31}$ | $15.60_{\pm 1.33}$ | $16.81_{\pm 0.72}$ | $7.27_{\pm 0.16}$ | $7.50_{\pm 0.01}$ |
| bbc | $\mathbf{34.66}_{\pm 0.15}$ | $238.77_{\pm 3.22}$ | $161.59_{\pm 10.8}$ | $236.07_{\pm 4.29}$ | $42.77_{\pm 0.75}$ | $35.02_{\pm 0.15}$ |
| bnetflix | $\mathbf{10.08}_{\pm 0.02}$ | $24.87_{\pm 0.64}$ | $20.53_{\pm 0.96}$ | $21.89_{\pm 0.34}$ | $10.56_{\pm 0.15}$ | $10.80_{\pm 0.00}$ |
| book | $\mathbf{3.65}_{\pm 0.02}$ | $121.44_{\pm 2.65}$ | $67.10_{\pm 9.40}$ | $95.96_{\pm 4.73}$ | $3.89_{\pm 0.18}$ | $3.87_{\pm 0.03}$ |
| c20ng | $\mathbf{19.12}_{\pm 0.04}$ | $215.58_{\pm 5.08}$ | $123.50_{\pm 5.85}$ | $187.31_{\pm 5.87}$ | $20.60_{\pm 0.26}$ | $20.34_{\pm 0.02}$ |
| cr52 | $\mathbf{11.94}_{\pm 0.10}$ | $201.12_{\pm 5.98}$ | $134.30_{\pm 7.81}$ | $217.42_{\pm 10.8}$ | $14.33_{\pm 0.34}$ | $12.75_{\pm 0.09}$ |
| cwebkb | $\mathbf{21.32}_{\pm 0.14}$ | $193.05_{\pm 3.82}$ | $109.07_{\pm 4.38}$ | $189.50_{\pm 7.03}$ | $24.40_{\pm 0.47}$ | $22.70_{\pm 0.07}$ |
| dna | $\mathbf{15.89}_{\pm 0.12}$ | $46.77_{\pm 1.38}$ | $28.76_{\pm 0.91}$ | $34.41_{\pm 1.21}$ | $21.19_{\pm 0.45}$ | $\mathbf{15.89}_{\pm 0.04}$ |
| jester | $\mathbf{9.04}_{\pm 0.02}$ | $24.39_{\pm 0.47}$ | $20.98_{\pm 0.33}$ | $21.08_{\pm 0.83}$ | $10.20_{\pm 0.12}$ | $10.66_{\pm 0.01}$ |
| kdd | $\mathbf{0.19}_{\pm 0.00}$ | $13.20_{\pm 1.73}$ | $4.12_{\pm 0.57}$ | $7.55_{\pm 1.30}$ | $0.33_{\pm 0.03}$ | $0.20_{\pm 0.00}$ |
| kosarek | $\mathbf{1.35}_{\pm 0.04}$ | $43.28_{\pm 2.04}$ | $13.57_{\pm 3.63}$ | $36.47_{\pm 3.56}$ | $2.15_{\pm 0.11}$ | $1.40_{\pm 0.00}$ |
| moviereview | $\mathbf{48.06}_{\pm 0.15}$ | $226.20_{\pm 4.90}$ | $141.19_{\pm 4.27}$ | $234.37_{\pm 10.9}$ | $60.11_{\pm 0.99}$ | $48.36_{\pm 0.06}$ |
| msnbc | $1.05_{\pm 0.03}$ | $2.68_{\pm 0.43}$ | $2.23_{\pm 0.39}$ | $2.12_{\pm 0.34}$ | $1.67_{\pm 0.01}$ | $\mathbf{0.99}_{\pm 0.00}$ |
| nltcs | $\mathbf{1.12}_{\pm 0.02}$ | $3.26_{\pm 0.27}$ | $2.48_{\pm 0.00}$ | $2.86_{\pm 0.39}$ | $1.66_{\pm 0.14}$ | $1.48_{\pm 0.00}$ |
| plants | $\mathbf{2.52}_{\pm 0.04}$ | $16.44_{\pm 1.20}$ | $11.62_{\pm 0.54}$ | $12.04_{\pm 0.40}$ | $3.22_{\pm 0.16}$ | $4.67_{\pm 0.01}$ |
| pumsb_star | $\mathbf{5.82}_{\pm 0.16}$ | $42.21_{\pm 1.94}$ | $23.30_{\pm 0.74}$ | $32.76_{\pm 1.53}$ | $8.99_{\pm 0.65}$ | $11.74_{\pm 0.02}$ |
| tmovie | $\mathbf{7.32}_{\pm 0.06}$ | $116.74_{\pm 1.89}$ | $57.94_{\pm 5.49}$ | $90.92_{\pm 10.4}$ | $10.39_{\pm 0.34}$ | $10.11_{\pm 0.02}$ |
| tretail | $\mathbf{1.22}_{\pm 0.01}$ | $30.08_{\pm 2.20}$ | $6.22_{\pm 1.63}$ | $34.79_{\pm 4.17}$ | $2.12_{\pm 0.25}$ | $\mathbf{1.22}_{\pm 0.00}$ |
| voting | $\mathbf{27.71}_{\pm 0.18}$ | $311.78_{\pm 3.31}$ | $264.53_{\pm 7.30}$ | $303.02_{\pm 13.6}$ | $91.81_{\pm 6.90}$ | $119.05_{\pm 0.11}$ |

APCs maintain the overall best performance across all datasets. However, we make three key observations. (1) Utilizing the encoding probabilistic circuit directly as a decoder ($APC_{pc}$) yields inferior performance across all tasks, with a median relative decrease of $7.3\times$ on tabular data $2.7\times$ in image data reconstruction, compared to employing a neural network decoder, highlighting, that the additional modeling capacity and flexibility of a neural decoder is crucial. (2) Integrating the SPAE scheme within the APC framework ($APC\text{-}SPAE_{pc}$) also led to a reduction in performance, further suggesting that a PC decoder is less effective in this context. (3) Incorporating SPAE-ACT and SPAE-CAT encoding mechanisms into the APC framework, while retaining a neural network decoder, consistently improved upon the performance of the baseline SPAE method in all evaluated scenarios, achieving median improvements of $2.5\times$ for ACT and $1.7\times$ for CAT embeddings on image data, and median $3.9\times$ and $4.8\times$ respectively on tabular datasets. We use median values here to reduce the impact of extreme outliers, which we obtained on some of the datasets.

We want to additionally highlight, that CAT embeddings [Vergari et al., 2018] can be interpreted as a special case of APCs. They are the result of running `MaxProdMPE` on the augmented circuit $\bar{\mathcal{C}}$ of $\mathbf{V} = (\mathbf{X}, \mathbf{Z}')$ where $\mathbf{Z}'$ are sum unit indicator input unit random variables obtained from the latent variable interpretation introduced in Peharz et al. [2017]. In contrast, the APC framework allows for embedding random variables to appear at *arbitrary positions* in the circuit graph represented by *arbitrary distributions*. This comparison highlights the flexibility of APCs compared to the prior autoencoding scheme introduced in Vergari et al. [2018].

# G KNOWLEDGE DISTILLATION: ALGORITHM & ADDITIONAL RESULTS

---

**Algorithm 2** Data-Free Knowledge Distillation from VAE to APC

---

**Require:** Pre-trained Teacher VAE, Student APC, Iterations $T$

1: **procedure** KNOWLEDGEDISTILLATION(VAE, APC, $T$)
2:     **for** $t = 1, 2, \ldots, T$ **do**
3:         $\mathbf{z} \sim p_{\mathcal{C}}(\mathbf{Z})$                                                       $\triangleright$ Sample embedding from APC prior
4:         $\hat{\mathbf{x}}_{\text{VAE}} = g_{\text{VAE}}(\mathbf{z})$                                  $\triangleright$ Generate synthetic data with VAE decoder
5:         $\mathbf{z}_{\text{VAE}} = f_{\text{VAE}}(\hat{\mathbf{x}}_{\text{VAE}})$                             $\triangleright$ Encode synthetic data with VAE encoder
6:         $\mathbf{z}_{\text{APC}} \sim p_{\mathcal{C}}(\mathbf{Z} \,|\, \mathbf{x}_{\text{VAE}})$                         $\triangleright$ Encode synthetic data with APC encoder
7:         $\hat{\mathbf{x}}_{\text{APC}} = g_{\text{APC}}(\mathbf{z}_{\text{APC}})$                                 $\triangleright$ Reconstruct with APC decoder
8:         $\mathcal{L} = \text{MSE}(\hat{\mathbf{x}}_{\text{VAE}}, \hat{\mathbf{x}}_{\text{APC}}) + \text{KLD}(p_{\mathcal{C}}(\mathbf{Z} \,|\, \hat{\mathbf{x}}_{\text{VAE}}) \,||\, \mathcal{N}(0,1)) - \log p_{\mathcal{C}}(\hat{\mathbf{x}}_{\text{VAE}}, \mathbf{z}_{\text{VAE}})$     $\triangleright$ Distillation loss
9:         Update APC parameters to minimize $\mathcal{L}$                             $\triangleright$ Gradient descent
10:     **return** Distilled APC

---

Appendix Algorithm 2 outlines the process of data-free knowledge distillation from a VAE to an APC of similar capacity and the same decoder architecture. The procedure iteratively refines the APC to learn from the VAE without need for original training data. At each iteration, an embedding $\mathbf{z}$ is first sampled from the APC prior $p_{\mathcal{C}}(\mathbf{Z})$. This embedding is then passed through the VAE decoder $g_{\text{VAE}}(\mathbf{z})$ to generate synthetic data $\hat{\mathbf{x}}_{\text{VAE}}$ that aligns with the APC prior. The synthetic data is subsequently re-encoded using the VAE encoder $\mathbf{z}_{\text{VAE}} = f_{\text{VAE}}(\hat{\mathbf{x}}_{\text{VAE}})$, to obtain an embedding according to the VAE's approximate posterior. In parallel, the APC encodes the same synthetic data using $p_{\mathcal{C}}(\mathbf{Z} \,|\, \mathbf{x}_{\text{VAE}})$, obtaining an auxiliary embedding $\mathbf{z}_{\text{APC}}$. This embedding is then passed through the APC decoder $g_{\text{APC}}(\mathbf{z}_{\text{APC}})$ to reconstruct the synthetic VAE sample, producing $\hat{\mathbf{x}}_{\text{APC}}$. The distillation loss $\mathcal{L}$ is equal to the main training recipe of APCs and consists of three components: the mean squared error (MSE) between the VAE and APC reconstructions, a Kullback-Leibler divergence (KLD) term that encourages the APC 's posterior $p_{\mathcal{C}}(\mathbf{Z} \,|\, \hat{\mathbf{x}}_{\text{VAE}})$ to align with a standard normal prior $\mathcal{N}(0,1)$, and a negative log-likelihood (NLL) term maximizing the joint likelihood of VAE sample and embedding $p_{\mathcal{C}}(\hat{\mathbf{x}}_{\text{VAE}}, \mathbf{z}_{\text{VAE}})$. Since all steps are differentiable, we can end-to-end train this procedure from Line 3 to Line 7 and update the APC parameters via gradient descent to minimize $\mathcal{L}$.

To demonstrate that knowledge distillation from VAEs to APCs generalizes beyond the two exemplary datasets considered in Appendix B.5, we extend our evaluation to all image and tabular datasets introduced in Appendix B. We report results for both teacher and student reconstructions, as well as downstream task accuracy under full evidence (Full Evi.) and MCAR-style corruptions, in Appendix Table 8.

Table 8: Data-free Knowledge Distillation. Input reconstruction mean squared error (↓) and downstream task accuracy (↑) performance comparison between VAE teacher and distilled APC student models, evaluated with full evidence (Full Evi.) and under varying levels of missing data (MCAR).

**Full Evidence**: APCs are successfully distilling the knowledge from their teacher.

**MCAR Corruptions**: APCs surpass their teacher in robustness against corruptions.

| | Teacher | → | Student |
|---|---|---|---|
| | | MSE (↓) | |
| MNIST | $3.15_{\pm0.02}$ | | $6.84_{\pm0.17}$ |
| F-MNIST | $5.84_{\pm0.02}$ | | $8.16_{\pm0.10}$ |
| CIFAR | $21.84_{\pm0.07}$ | | $25.34_{\pm0.30}$ |
| CelebA | $336.00_{\pm22.7}$ | | $402.53_{\pm21.1}$ |
| Flowers | $115.04_{\pm2.12}$ | | $57.66_{\pm1.33}$ |
| LSUN | $104.91_{\pm14.3}$ | | $119.07_{\pm11.1}$ |
| SVHN | $15.04_{\pm0.48}$ | | $13.14_{\pm0.50}$ |
| ImageNet | $1375.88_{\pm30.5}$ | | $230.16_{\pm11.3}$ |
| accidents | $10.38_{\pm0.67}$ | | $19.03_{\pm1.79}$ |
| ad | $3.97_{\pm0.22}$ | | $11.75_{\pm0.84}$ |
| baudio | $11.47_{\pm0.18}$ | | $20.35_{\pm2.95}$ |
| bbc | $71.49_{\pm0.55}$ | | $80.06_{\pm3.29}$ |
| bnetflix | $17.61_{\pm0.24}$ | | $21.59_{\pm3.31}$ |
| book | $7.47_{\pm0.06}$ | | $13.09_{\pm3.13}$ |
| c20ng | $38.61_{\pm0.03}$ | | $64.73_{\pm10.3}$ |
| cr52 | $22.66_{\pm0.21}$ | | $31.68_{\pm2.27}$ |
| cwebkb | $43.63_{\pm0.36}$ | | $57.91_{\pm3.21}$ |
| dna | $32.34_{\pm0.07}$ | | $40.59_{\pm3.39}$ |
| jester | $16.38_{\pm0.23}$ | | $25.35_{\pm2.74}$ |
| kdd | $0.37_{\pm0.00}$ | | $0.43_{\pm0.04}$ |
| kosarek | $2.36_{\pm0.03}$ | | $2.96_{\pm0.15}$ |
| moviereview | $102.20_{\pm0.80}$ | | $131.58_{\pm13.0}$ |
| msnbc | $1.58_{\pm0.09}$ | | $1.96_{\pm0.02}$ |
| nltcs | $1.05_{\pm0.00}$ | | $1.32_{\pm0.02}$ |
| plants | $2.65_{\pm0.23}$ | | $3.55_{\pm0.37}$ |
| pumsb_star | $8.14_{\pm0.70}$ | | $20.08_{\pm5.77}$ |
| tmovie | $13.15_{\pm0.03}$ | | $19.06_{\pm1.22}$ |
| tretail | $2.20_{\pm0.00}$ | | $2.54_{\pm0.06}$ |
| voting | $51.00_{\pm0.97}$ | | $57.25_{\pm2.31}$ |
| | | Downstream Accuracy (↑) | |
| MNIST | $96.12_{\pm0.12}$ | | $90.39_{\pm0.07}$ |
| F-MNIST | $83.02_{\pm0.16}$ | | $81.04_{\pm0.21}$ |
| SVHN | $56.92_{\pm3.78}$ | | $23.98_{\pm0.35}$ |
| CIFAR | $47.14_{\pm0.37}$ | | $38.62_{\pm0.42}$ |
| Flowers | $9.87_{\pm0.65}$ | | $12.84_{\pm0.23}$ |
| ImageNet | $6.60_{\pm0.11}$ | | $2.97_{\pm0.14}$ |

| | Teacher | → | Student |
|---|---|---|---|
| | | MSE (↓) | |
| MNIST | $35.38_{\pm1.74}$ | | $6.15_{\pm0.11}$ |
| F-MNIST | $39.80_{\pm1.10}$ | | $5.33_{\pm0.05}$ |
| CIFAR | $55.81_{\pm0.47}$ | | $19.61_{\pm0.10}$ |
| CelebA | $1095.90_{\pm14.0}$ | | $201.70_{\pm10.4}$ |
| Flowers | $107.75_{\pm11.8}$ | | $31.43_{\pm0.51}$ |
| LSUN | $254.75_{\pm2.39}$ | | $68.71_{\pm4.23}$ |
| SVHN | $41.01_{\pm0.15}$ | | $7.68_{\pm0.19}$ |
| ImageNet | $1286.51_{\pm10.4}$ | | $139.72_{\pm6.39}$ |
| accidents | $7.56_{\pm0.28}$ | | $7.76_{\pm0.37}$ |
| ad | $6.04_{\pm0.17}$ | | $5.68_{\pm0.23}$ |
| baudio | $7.44_{\pm0.01}$ | | $7.94_{\pm0.40}$ |
| bbc | $37.45_{\pm0.55}$ | | $39.04_{\pm1.20}$ |
| bnetflix | $13.49_{\pm0.77}$ | | $10.46_{\pm0.41}$ |
| book | $3.73_{\pm0.02}$ | | $6.12_{\pm0.85}$ |
| c20ng | $20.69_{\pm0.05}$ | | $29.86_{\pm1.85}$ |
| cr52 | $12.83_{\pm0.18}$ | | $15.98_{\pm0.61}$ |
| cwebkb | $22.94_{\pm0.15}$ | | $28.13_{\pm1.24}$ |
| dna | $16.46_{\pm0.17}$ | | $19.29_{\pm1.09}$ |
| jester | $17.44_{\pm0.10}$ | | $10.37_{\pm0.43}$ |
| kdd | $0.19_{\pm0.00}$ | | $0.22_{\pm0.01}$ |
| kosarek | $1.42_{\pm0.01}$ | | $1.58_{\pm0.05}$ |
| moviereview | $50.08_{\pm0.22}$ | | $62.55_{\pm3.66}$ |
| msnbc | $1.03_{\pm0.01}$ | | $1.07_{\pm0.04}$ |
| nltcs | $1.60_{\pm0.01}$ | | $1.12_{\pm0.01}$ |
| plants | $3.93_{\pm0.05}$ | | $2.55_{\pm0.05}$ |
| pumsb_star | $12.47_{\pm0.40}$ | | $9.22_{\pm2.80}$ |
| tmovie | $8.72_{\pm0.03}$ | | $8.81_{\pm0.33}$ |
| tretail | $1.23_{\pm0.00}$ | | $1.22_{\pm0.01}$ |
| voting | $122.24_{\pm52.8}$ | | $28.94_{\pm0.82}$ |
| | | Downstream Accuracy (↑) | |
| MNIST | $27.49_{\pm2.23}$ | | $87.77_{\pm0.12}$ |
| F-MNIST | $30.30_{\pm2.64}$ | | $79.41_{\pm0.24}$ |
| SVHN | $36.38_{\pm1.99}$ | | $22.93_{\pm0.29}$ |
| CIFAR | $33.53_{\pm0.50}$ | | $37.66_{\pm0.33}$ |
| Flowers | $4.53_{\pm0.35}$ | | $12.56_{\pm0.27}$ |
| ImageNet | $5.54_{\pm0.37}$ | | $2.97_{\pm0.15}$ |

## H EMBEDDING-BASED OUT-OF-DISTRIBUTION DETECTION USING APCS

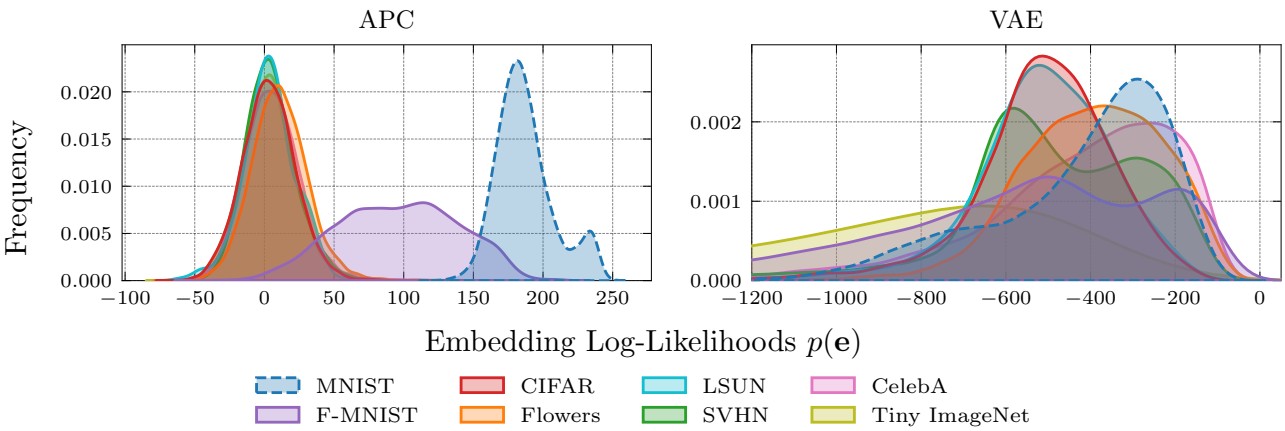

Figure 16: APC and VAE models under the lens of out-of-distribution data: Models were trained on MNIST and embedding log-likelihoods $p(\mathbf{z})$ were evaluated for a range of out-of-distribution datasets. For APCs we can tractably obtain $p(\mathbf{z}) = \int p(\mathbf{x}, \mathbf{z}) \, d\mathbf{x}$ with marginalization. For VAEs, we need to compute an aggregate posterior from training dataset embeddings. While VAEs map in-distribution and out-of-distribution data to the same embedding range, APCs are able to cleanly separate MNIST from other datasets. As expected, we see some minor overlap with F-MNIST due to their similar pixel statistics, whereas all other datasets have close to no overlap with MNIST.

In addition to generative tasks, we can also investigate the empirical distribution of embedding likelihoods produced by the model. Such analysis could potentially allow out-of-distribution (OOD) detection, enabling decoding processes or downstream applications to proactively *reject* embeddings that deviate significantly from the expected embedding distribution, as determined by a predefined threshold. As an initial exploration, Appendix Figure 16 compares the embedding log-likelihood distributions of APCs and VAEs for in-distribution (MNIST) and out-of-distribution datasets. We selected MNIST for our OOD experiments because its hand-drawn digit images represent a highly constrained and artificial distribution that differs from naturally occurring images. This distinctive characteristic, which MNIST shares with F-MNIST, establishes a clear distributional boundary that should theoretically enable models to assign lower likelihood scores to out-of-distribution samples compared to in-distribution data points. As previously established, APCs allow us to evaluate the marginal embedding log-likelihood exact and tractably, while VAEs require approximating this distribution using a post-hoc aggregate posterior from training data. As becomes evident in Appendix Figure 16, APC embeddings separate in-distribution from out-of-distribution data in the likelihood space, with minimal overlap except for F-MNIST, which shares similar pixel statistics with MNIST. In contrast, VAE embeddings exhibit substantial overlap between in-distribution and out-of-distribution likelihood scores, making OOD detection almost impossible. This demonstrates that APCs' explicit probabilistic formulation provides not only strong reconstruction and representation learning capabilities but also hints at out-of-distribution detection capabilities without requiring additional mechanisms or models. We leave further in-depth analysis on this topic to future work.

# I  DIFFERENTIABLE SAMPLING WITH SIMPLE

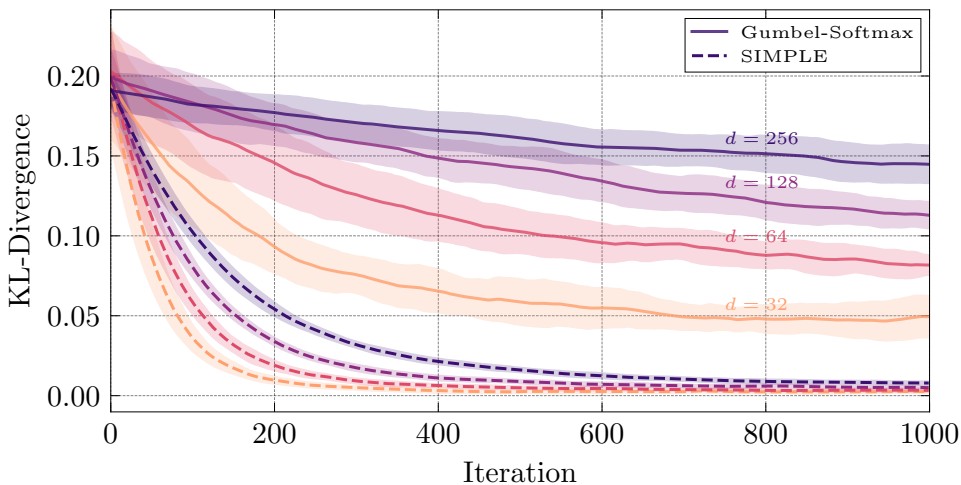

Figure 17: **SIMPLE outperforms Gumbel-Softmax in gradient estimation** as measured by the KLD between ground-truth and learned sum units for different input dimensions $d$.

---

**Algorithm 3** Sampling sum units with SIMPLE

---

**Require:** Normalized weights $\boldsymbol{\theta} \in [0,1]^D$ for sum unit $n$, where $D = |\text{IN}(n)|$.
1: **procedure** SIMPLE($\boldsymbol{\theta}$)
2:     $\log \boldsymbol{\theta} \leftarrow \log(\boldsymbol{\theta})$                                                                    ▷ Convert probabilities to log-probabilities (logits)
3:     Draw $\mathbf{g} \sim \text{Gumbel}(0,1)^D$                                                       ▷ Draw independent Gumbel noise for each component
4:     $\mathbf{z} \leftarrow \log \boldsymbol{\theta} + \mathbf{g}$                                                          ▷ Perturb log-probabilities with Gumbel noise
5:     $d^* \leftarrow \arg\max_{j=1,\ldots,D} z_j$                                   ▷ Identify the index of the maximum perturbed logit (discrete sample)
6:     $\mathbf{s} \leftarrow \text{one-hot}(d^*, D)$                                                     ▷ Construct a one-hot vector $\mathbf{s}$ where $s_{d^*} = 1$
7:     **return** $(\mathbf{s} - \boldsymbol{\theta}).detach() + \boldsymbol{\theta}$                             ▷ Return $\mathbf{s}$ for forward pass, but only pass gradients of $\boldsymbol{\theta}$ (*detach*)

---

We additionally improve the differentiable sampling approach for PCs originally proposed in Lang et al. [2022] by replacing the Gumbel-Softmax gradient estimator with SIMPLE [Ahmed et al., 2023] for $k$-subset sampling (with $k = 1$) when sampling from sum units as outlined in Appendix Algorithm 3. In contrast to Gumbel-Softmax, SIMPLE directly propagates the gradients through the unperturbed paramters $\boldsymbol{\theta}$. To quantitatively evaluate this modification, we conducted a controlled experiment comparing both gradient estimators on a synthetic task: learning a sum unit distribution with 32,64,128, and 256 inputs to match a known categorical ground truth distribution over 64 categories. We optimized sum unit weights by minimizing the MSE between one-hot encoded samples from both distributions, using AdamW with a learning rate of 0.01 for 1,000 iterations with batch size 64, repeating across 10 random seeds. As shown in Appendix Figure 17, SIMPLE demonstrates faster convergence and higher accuracy in approximating the true distribution, as measured by KLD. Notably, SIMPLE at iteration 100 achieves comparable performance to Gumbel-Softmax at iteration 1,000, and continues to improve beyond that point. Furthermore, SIMPLE demonstrates lower variance across different random initializations, indicating more consistent convergence behavior regardless of starting conditions. Upon convergence, SIMPLE achieves a KLD of $0.0033 \pm 0.00065$, approximately $25\times$ lower than Gumbel-Softmax's $0.0817 \pm 0.00602$ for $D = 64$. These results validate our choice of SIMPLE as the gradient estimator for differentiable sampling in APCs, enabling more accurate and stable training.

# J  FUTURE WORK

While our study has thoroughly examined the APC framework, numerous avenues remain for further research. Using a circuit-based encoder grants explicit control over both data and embedding distributions, lifting the typical restriction to

independent Gaussians, as seen in VAEs. Moreover, the circuit structures employed in this work were selected heuristically – EinsumNetworks for tabular data and a convolutional structure for image data. However, the broader literature on PCs offers various approaches for learning circuit structures. Incorporating these methods could enable more strategic placement of embedding input units, potentially distributing them across different circuit levels. This could facilitate the formation of low-level and high-level representations, introducing hierarchical structures in the learned embedding space. Furthermore, Appendix B.4 shows that while modern VAEs achieve high-quality reconstructions on complete data using novel scaling and training tricks, they remain vulnerable to corruption. Future work could thus apply analogous methodologies to scale PC encoders, aiming for NVAE-level reconstruction fidelity on complete data while preserving APCs' inherent robustness to data corruptions.