# OpenReview forum: "Autoencoding Probabilistic Circuits"
_auai.org/UAI/2025/Workshop/TPM — TPM 2025_

### Official Review · Reviewer_zKje · 2025-06-10

**Rating:** 2

**Review:**

The paper proposes using PCs as encoders in a autoencoder framework as a means to enable exact inference over the latent variables $z$ even when the data $x$ is only partially observed. In contrast to previous work by Vergari et al., the authors propose to model the latent variables as explicit nodes in the PC graphs, which are jointly learned with the rest of the encoder architecture. This learned representation seem to be effective at enabling accurate reconstructions and downstream tasks under missing data, outperforming previous methods.

### Strengths

The proposed method cleverly exploits the exact inference capabilities of PCs to handle missing data in the encoding process of autoencoders, while still using a more expressive neural network for decoding. The expression of the KL in exact form in the space of PC graphs is also quite elegant. In terms of empirical results, the experiments were extensive and well designed, albeit with some missing details (see question 3 and 4). The distillation idea is particularly interesting and potentially very useful in a number of applications.

### Weaknesses

Some important details are not clear or discussed in the paper. See questions below.

### Questions
1. Could you elaborate further on the training procedure? In principle, since the latent variable $z$ is not observed, one would typically learn $p(x)$ by marginalizing $z$ out. That's the approach with PCs (where the latent variables correspond to sum nodes) and VAEs, where the loss is given by the sum of the reconstruction and KL terms which form an lower bound to $p(x)$. That is why I am somewhat confused by the need of the joint term in (5).
    - Would it not suffice to optimize (4) and (6) as in VAEs?
    - What are the $z_i$ values that we optimize over in (5), as $z$ is not observed?
    - How are the loss terms combined? Is the total loss a simple sum or are there coefficients controlling the influence of each term?

2. Since training requires computing the marginal $p(x)$ to get the conditional $p(z|x)$, would it be helpful to pretrain the encoder to fit $p(x)$ via MLE as is done in regular PCs? I imagine this could speed up convergence.

3. The design of the encoder architecture is not entirely clear. Where should the latent variables be placed in the graph? This probably has an important effect on the expressivity of the encoder. One could naively construct an architecture where $p(x,z) = p(x)p(z)$ where the latents are not useful.

4. It would be useful to know more about the training details. Unless I missed it, the paper does not elaborate on how the data is presented to the model during training. What is the rate of missing data during training? Is it the same for all models, e.g., MIWAE?

---

### Official Review · Reviewer_E4Fq · 2025-06-13
**PCs with neural decoders for tractable representation learning; well written and motivated; thorough experiments**

**Rating:** 3

**Review:**

This paper presents Autoencoding Probabilistic Circuits, a hybrid framework that combines the tractable, exact inference capabilities of Probabilistic Circuits with the representational flexibility of neural networks for improved encoding in autoencoding tasks. By explicitly modeling the joint distribution over data and embeddings, APCs support exact conditional sampling and marginalization, enabling robust end-to-end training even in the presence of substantial missing data. The empirical evaluations are extensive and thorough, demonstrating consistent advantages across reconstruction quality, downstream classification, out-of-distribution detection.

While the use of a neural decoder introduces some opacity in the generative process and the framework does not currently provide uncertainty estimates over reconstructions, these are understandable design choices and do not detract from the core contributions. That said, I encourage the authors to consider comparisons to normalizing flows, which similarly provide invertible latent mappings. Additionally, the discussion around SPAE-based baselines could be expanded to better articulate the limitations and better position the paper.